# GRAPE🍇: Optimize Data Mixture
# for Group Robust Multi-target Adaptive Pretraining

**Simin Fan, Maria Ios Glarou, Martin Jaggi**
MLO, EPFL
`firstname.lastname@epfl.ch`

## Abstract

The performance of large language models (LLMs) across diverse downstream applications is fundamentally governed by the quality and composition of their pretraining corpora. Existing domain reweighting algorithms primarily optimize data mixtures for a single target task, thereby resulting in models that overfit to specialized objectives while exhibiting substantial performance degradation on other benchmarks. This paper introduces **G**roup **R**obust Multi-target **A**daptive **PrE**training (GRAPE), a novel multi-source-multi-target domain reweighting framework designed to calibrate pretraining data mixtures for robust performance across multiple target tasks simultaneously. GRAPE dynamically adjusts sampling weights across source domains (*domain weights*) while concurrently modulating *task weights* that quantify the relative importance of each individual target task. This adaptive process prioritizes tasks based on their learning difficulty throughout training. We formulate this interleaved reweighting mechanism as a minimax optimization problem: The inner maximization adjusts task weights leveraging group distributed-robust-optimization (DRO), where those tasks demonstrating the least improvement under the current data mixture are prioritized with higher weights; The outer minimization then optimizes domain weights to maximize loss reduction on the prioritized tasks. Experiments on `ClimbLab` and `SlimPajama` datasets demonstrate that GRAPE consistently outperforms baseline methods in terms of reasoning performance across 6 benchmarks. Furthermore, when applied to multilingual targets, GRAPE effectively identifies optimal training mixtures from mainstream languages, achieving superior language modeling capabilities across 8 low-resource target languages.

## 1 Introduction

The composition of pretraining data, typically encompassing diverse sources, topics, and languages [Gao et al., 2020, Together Computer, 2023], is a critical determinant of large language model (LLM) capabilities. While various data sources are conventionally combined using fixed, heuristically determined mixing ratios, recent research indicates that adaptively adjusting these ratios (*domain weights*) via **domain reweighting** during pretraining can substantially enhance downstream task performance [Xie et al., 2023, Liu et al., 2025, Fan et al., 2024a]. However, prevailing domain reweighting methods are largely confined to the *multi-source-single-target* paradigm, focusing exclusively on source-target relevance while failing to address complex interactions across multiple target tasks. In particular, task-adaptive domain reweighting approaches [Grangier et al., 2025, Fan et al., 2024b] curate data mixtures based on performance on one specific task, while often leading to overfitting with compromised generalization ability. General domain reweighting techniques such as Liu et al. [2025] and Fan et al. [2024a] adjust training data mixtures for average performance across various target tasks, assessed via mean stochastic task losses or gradients. Yet, due to the heterogeneous dynamics of multi-task learning, average assessment proves suboptimal when confronted with dominant gradients – where loss or gradient magnitude on certain tasks overshadow others, or conflicting gradients – where improving one task hinders another. Conversely, traditional multi-task learning (MTL) strategies like gradient surgery [Yu et al., 2020, Liu et al., 2023, Bohn et al.,

2024] require substantial data from each target task and often involve complex gradient manipulations that scale poorly to foundation models, rendering them infeasible for LLM pretraining.

To facilitate effective multi-task adaptation during LLM pretraining, we propose **G**roup **R**obust Multi-target **A**daptive **PrE**training (GRAPE), the first domain reweighting algorithm explicitly designed for the *multi-source-multi-target* setting. Instead of optimizing for average task performance, GRAPE calibrates domain weights to maximize improvement on the most challenging tasks at each stage of training, guided by the key principle:

*Tasks that are improving slowly deserve more attention.*

Specifically, GRAPE employs group distributed-robust-optimization (DRO) [Duchi and Namkoong, 2020, Sagawa et al., 2020] to adaptively assign higher *task weights* to target tasks that are currently learning slowest. We introduce the *Rate-of-Improvement* (*RoI*, Equation 1), a normalized, step-wise metric assessing task-specific learning progress. Unlike prior metrics like excess loss [Xie et al., 2023], RoI avoids reliance on auxiliary models and inherently accounts for varying task difficulties by normalizing improvement against current performance.

We formulate this bifold reweighting process as a *minimax* optimization problem (Equation 2), jointly optimizing: (1) **domain weights** ($\boldsymbol{\alpha}$), the sampling distribution over source domains; and (2) **task weights** ($\boldsymbol{z}$), a distribution over target tasks reflecting their priority during the evolution of training. The inner maximization phase uses DRO to identify the task distribution $\boldsymbol{z}$ that emphasizes the least-improved tasks (minimal RoI). Tasks demonstrating minimal RoI under the current domain mixture are thus prioritized in subsequent optimization steps; The outer minimization phase then optimizes the domain weights $\boldsymbol{\alpha}$ to maximize the expected improvement according to this dynamically determined worst-case task distribution. This interleaved mechanism enables GRAPE to balance domain utilization against multi-task performance adaptively via a dynamic curriculum, preventing dominance by any single task and fostering robust generalization.

**Contributions.** Our primary contributions are threefold:

• We introduce GRAPE, the first principled *multi-target domain reweighting* algorithm for LLM pretraining. We provide a rigorous derivation grounded in optimization theory, including a theoretical analysis of its properties (§ 2).

• We empirically validate GRAPE's efficacy, demonstrating superior performance over baseline methods on reasoning benchmarks (§ 3.1) and low-resource multilingual language modeling (§ 3.2).

• Through comprehensive ablation studies and comparisons of different progress assessment metrics used in DRO, we offer insights on the training dynamics of multi-target domain reweighting, shed light on the future research in data-centric LLM pretraining and curriculum learning (§ 4).

## 2 Methodology: Multi-Target Domain Reweighting via GRAPE

This section formalizes the problem of domain reweighting for multi-target adaptation and presents GRAPE, our proposed algorithm that optimizes pretraining data composition for robust performance across multiple target tasks.

**Problem Formulation.** Let $\mathcal{D}_{\text{train}} \triangleq \{\mathcal{D}_1, \ldots, \mathcal{D}_K\}$ represent a pretraining corpus partitioned into $k$ source domains based on meta-attributes (e.g., source, topic, etc.). During training, data is sampled from $\mathcal{D}_{\text{train}}$ according to *domain weights* $\boldsymbol{\alpha} \in \Delta^K$, where $\Delta^K$ is the probability simplex in $\mathbb{R}^K$. Let $\mathcal{D}_{\text{tgt}} \triangleq \{\mathcal{T}_1, \ldots, \mathcal{T}_N\}$ be a set of $N$ target tasks, which can be presented as different benchmarks. Our objective is to find the optimal domain weights $\boldsymbol{\alpha}$ that yield strong performance across all tasks $\mathcal{T}_n \in \mathcal{D}_{\text{tgt}}$. We introduce *task weights* $\boldsymbol{z} \in \Delta^N \subset \mathbb{R}^N$ to represent the relative importance assigned to each target task during training. The core idea is to dynamically update $\boldsymbol{z}$ to reflect task learning progress and subsequently adjust $\boldsymbol{\alpha}$ to optimize the training trajectory based on $\boldsymbol{z}$.

Let $\boldsymbol{\theta}_t$ denote the model parameters at step $t$, and let $l_n(\boldsymbol{\theta}_t)$ be the stochastic loss evaluated on target task $\mathcal{T}_n$ for $n \in [N]$. We assume optimization proceeds via gradient descent: $\boldsymbol{\theta}_{t+1} = \boldsymbol{\theta}_t - \gamma_t \boldsymbol{d}_t$, where $\gamma_t$ is the learning rate and $\boldsymbol{d}_t$ is the update direction. Let $l_k(\boldsymbol{\theta}_t), g_k(\boldsymbol{\theta}_t)$ be the stochastic loss and gradient computed on data from training domain $\mathcal{D}_k$ for $k \in [K]$. The overall parameter update direction is a weighted combination of stochastic domain gradients: $\boldsymbol{d}_t := \sum_{k=1}^{K} \alpha_k^t g_k(\boldsymbol{\theta}_t)$, where the domain weights $\boldsymbol{\alpha}^t$ are used as the dynamic importance sampling weights at step $t$.

## 2.1 GRAPE: Prioritizing the Lagged Tasks by Group DRO

**Algorithm.** To achieve balanced improvement across target tasks, GRAPE employs Group Distributed Robust Optimization (Group DRO) to jointly adapt task weights ($z$) and domain weights ($\alpha$). We assess task improvement using the *Rate-of-Improvement (RoI)*, inspired by Liu et al. [2023]:

$$r_n^{(t)} := \frac{l_n(\boldsymbol{\theta}_t) - l_n(\boldsymbol{\theta}_{t+1})}{l_n(\boldsymbol{\theta}_t)}, \tag{1}$$

Here, $l_n(\boldsymbol{\theta}_t) - l_n(\boldsymbol{\theta}_{t+1})$ approximates the one-step improvement in loss for task $\mathcal{T}_n$. Normalizing by $l_n(\boldsymbol{\theta}_t)$ provides a scale-invariant measure of relative progress, preventing bias towards tasks with intrinsically higher loss magnitudes. For sufficiently small learning rates $\gamma_t$, a first-order Taylor expansion yields:

$$r_n^{(t)} = \frac{\langle \nabla_{\boldsymbol{\theta}} l_n(\boldsymbol{\theta}_t), \gamma_t \boldsymbol{d}_t \rangle + o(\|\boldsymbol{d}_t\|^2)}{l_n(\boldsymbol{\theta}_t)} \approx \gamma_t \langle \nabla_{\boldsymbol{\theta}} \log l_n(\boldsymbol{\theta}_t), \boldsymbol{d}_t \rangle = \gamma_t \sum_{k=1}^{K} \alpha_k^t \langle \nabla_{\boldsymbol{\theta}} \log l_n(\boldsymbol{\theta}_t), \boldsymbol{g}_k(\boldsymbol{\theta}_t) \rangle,$$

where $\nabla_{\boldsymbol{\theta}} \log l_n(\boldsymbol{\theta}_t) = \frac{\nabla_{\boldsymbol{\theta}} l_n(\boldsymbol{\theta}_t)}{l_n(\boldsymbol{\theta}_t)}$. Following the principles of Group DRO [Sagawa et al., 2020, Qi et al., 2021], we formulate a minimax objective to optimize the domain weights ($\alpha$) for the worst-case (minimum) RoI score across tasks:

$$\max_{\boldsymbol{\alpha} \in \Delta^K} \min_{n \in [N]} \mathbb{E}[r_n^{(t)}] \approx \max_{\boldsymbol{\alpha} \in \Delta^K} \min_{\boldsymbol{z} \in \Delta^N} \sum_{n=1}^{N} z_n \mathbb{E}[r_n^{(t)}] - \overline{h}(\boldsymbol{\alpha}) + \underline{h}(\boldsymbol{z}) \tag{2}$$

$$\approx \max_{\boldsymbol{\alpha} \in \Delta^k} \min_{\boldsymbol{z} \in \Delta^N} \gamma_t \sum_{k=1}^{K} \alpha_k \sum_{n=1}^{N} z_n \mathbb{E}[\langle \nabla_{\boldsymbol{\theta}} \log l_n(\boldsymbol{\theta}_t), \boldsymbol{g}_k(\boldsymbol{\theta}_t) \rangle] - \overline{h}(\boldsymbol{\alpha}) + \underline{h}(\boldsymbol{z}).$$

Here, the inner minimization finds the task weights $z$ that concentrate on the tasks with the lowest expected RoI; The outer maximization finds the domain weights $\alpha$ that maximize this minimum expected RoI. Following Qi et al. [2021], Namkoong and Duchi [2016], we introduce Bregman divergence regularization terms $\overline{h}(\boldsymbol{\alpha})$ and $\underline{h}(\boldsymbol{z})$ relative to the previous weights $\boldsymbol{\alpha}^{(t-1)}$ and $\boldsymbol{z}^{(t-1)}$ to stabilize the updates and manage stochastic variance:

$$\overline{h}(\boldsymbol{\alpha}) := \mu_{\boldsymbol{\alpha}} D_{\Psi}(\boldsymbol{\alpha} \| \boldsymbol{\alpha}^{(t-1)}), \quad \underline{h}(\boldsymbol{z}) := \mu_{\boldsymbol{z}} D_{\Psi}(\boldsymbol{z} \| \boldsymbol{z}^{(t-1)}),$$

where $\Psi(\boldsymbol{b}) = \sum_i b_i \log(b_i)$ is the negative entropy. Solving this regularized minimax problem (details in Appendix B) yields multiplicative update rules resembling online mirror descent:

$$\boldsymbol{z}^t = \frac{\hat{\boldsymbol{z}}^t}{\sum_{n \in [N]} \hat{z}_n^t}, \text{ with } \hat{z}_n^t \leftarrow z_n^{t-1} \cdot \exp\left( -\frac{\gamma_t}{\mu_{\boldsymbol{z}}} \mathbb{E}_{\boldsymbol{x} \sim \text{mix}(\boldsymbol{\alpha}^{t-1})} [\langle \nabla_{\boldsymbol{\theta}} \log l_n(\boldsymbol{\theta}_t), \nabla_{\boldsymbol{\theta}} \ell(\boldsymbol{\theta}_t; \boldsymbol{x}) \rangle] \right), \tag{3}$$

$$\boldsymbol{\alpha}^t = \frac{\hat{\boldsymbol{\alpha}}^t}{\sum_{k \in [K]} \hat{\alpha}_k^t}, \text{ with } \hat{\alpha}_k^t \leftarrow \alpha_k^{t-1} \cdot \exp\left( \frac{\gamma_t}{\mu_{\boldsymbol{\alpha}}} \mathbb{E}_{\boldsymbol{y} \sim \text{mix}(\boldsymbol{z}^{t-1})} [\langle \boldsymbol{g}_k(\boldsymbol{\theta}_t), \nabla_{\boldsymbol{\theta}} \log \ell(\boldsymbol{\theta}_t; \boldsymbol{y}) \rangle] \right). \tag{4}$$

In practice, the expectations are replaced by stochastic estimates using mini-batches. The inner product $\langle \nabla_{\boldsymbol{\theta}} \log l_n(\boldsymbol{\theta}_t), \boldsymbol{g}_k(\boldsymbol{\theta}_t) \rangle$ captures the alignment between the gradient direction of task $n \in [N]$ (normalized by loss) and the gradient direction of domain $k \in [K]$. $\nabla_{\boldsymbol{\theta}} \ell(\boldsymbol{\theta}_t; \boldsymbol{x})$ denotes the (training loss) gradient from a training batch $\boldsymbol{x}$ sampled according to $\boldsymbol{\alpha}^{t-1}$, and $\nabla_{\boldsymbol{\theta}} \log \ell(\boldsymbol{\theta}_t; \boldsymbol{y})$ denotes the task-weighted normalized validation gradient from a validation batch $\boldsymbol{y}$ sampled according to $\boldsymbol{z}^{t-1}$. The pseudocode is presented in Algorithm 1.

**Efficiency analysis.** Updating $\alpha$ and $z$ at every step (Equation 3, Equation 4) requires $N$ target gradients and $K$ domain gradients computations, which is computationally expensive. We mitigate this by performing updates periodically, where domain weights ($\alpha$) and task weights ($z$) are updated every $\Delta T_{\boldsymbol{z}}$ and $\Delta T_{\boldsymbol{\alpha}}$ steps. For our experiments on `SlimPajama` with $K$=7 and $N$=6, using $\Delta T_{\boldsymbol{z}} = 100, \Delta T_{\boldsymbol{\alpha}} = 100$, the computational overhead is approximately $(N+1)\Delta T_z^{-1} + (K+1)\Delta T_{\alpha}^{-1}$=15% in terms of gradient computations relative to standard training. Compared to prior domain reweighting algorithm DOGE [Fan et al., 2024a], our task reweighting step only introduce $(N+1)\Delta T_z^{-1}$=7% overhead in computations. The memory overhead stems primarily from storing gradients for each domain/task during the update step. If model parameters require $m$ storage, `AdamW` optimizer stores

---
**Algorithm 1** **G**roup **R**obust **M**ulti-target **A**daptive **Pr**Etraining (GRAPE)
---
1: **Input:** Training domains $\mathcal{D}_{\text{train}} \triangleq \{\mathcal{D}_1, \ldots, \mathcal{D}_K\}$, target task validation sets $\mathcal{D}_{\text{tgt}} \triangleq \{\mathcal{T}_1, \ldots, \mathcal{T}_N\}$, optimizer `Optimizer`, loss function $l(\cdot)$, initial parameters $\boldsymbol{\theta}^0$, learning rate schedule $\gamma_0$, initial weights $\boldsymbol{\alpha}^0, \boldsymbol{z}^0$, update frequencies $\Delta T_{\boldsymbol{\alpha}}, \Delta T_{\boldsymbol{z}}$, regularization coefficients $\mu_{\boldsymbol{\alpha}}, \mu_{\boldsymbol{z}}$, total steps $T$.
2: **for** $t \in \{0 \ldots T\}$ **do**                                       # *Standard training: update model parameters $\boldsymbol{\theta}$*
3:         Sample a batch from $\mathcal{D}_{\text{train}}$: $\boldsymbol{x} \sim \text{mix}(\boldsymbol{\alpha}^t)$
4:         Update model parameters $\boldsymbol{\theta}^{t+1} \leftarrow \texttt{Optimizer}(\boldsymbol{\theta}^t, \nabla_{\boldsymbol{\theta}} \ell(\boldsymbol{\theta}^t, \boldsymbol{x}))$
5:         **if** $t\%\Delta T_{\boldsymbol{z}} = 0$ **then**                              # *Task Reweighting: update task weights $\boldsymbol{z}$*
6:                 Sample one batch from each target task: $\boldsymbol{y}_n \sim \mathcal{T}_n \in \mathcal{D}_{\text{tgt}}$ for $n \in [N]$
7:                 Sample one batch from $\mathcal{D}_{\text{train}}$: $\boldsymbol{x} \sim \text{mix}(\boldsymbol{\alpha}^t)$
8:                 Compute gradient alignments: $\boldsymbol{a}_n^t \leftarrow \langle \nabla_{\boldsymbol{\theta}} \log \ell(\boldsymbol{\theta}^{t+1}; \boldsymbol{y}_n), \nabla_{\boldsymbol{\theta}} \ell(\boldsymbol{\theta}^{t+1}; \boldsymbol{x}) \rangle$
9:                 Update *task weights*: $\boldsymbol{z}^{t+1} \leftarrow \dfrac{\hat{\boldsymbol{z}}}{\sum_{n=1}^N \hat{\boldsymbol{z}}_n}$ with $\hat{\boldsymbol{z}} = \boldsymbol{z}^t \odot \exp\left(-\dfrac{\gamma_t}{\mu_{\boldsymbol{z}}} \boldsymbol{a}^t\right)$
10:        **else**        $\boldsymbol{z}^{t+1} \leftarrow \boldsymbol{z}^t$
11:        **end if**
12:        **if** $t\%\Delta T_{\boldsymbol{\alpha}} = 0$ **then**                              # *Domain Reweighting: update domain weights $\boldsymbol{\alpha}$*
13:                Sample one batch from each domain: $\boldsymbol{x}_k \sim \mathcal{D}_k \in \mathcal{D}_{\text{train}}$ for $k \in [K]$
14:                Sample one batch from $\mathcal{D}_{\text{tgt}}$: $\boldsymbol{y} \sim \text{mix}(\boldsymbol{z}^t)$
15:                Compute gradient alignments: $\boldsymbol{a}_k^t \leftarrow \langle \nabla_{\boldsymbol{\theta}} \ell(\boldsymbol{\theta}^{t+1}; \boldsymbol{x}_k), \nabla_{\boldsymbol{\theta}} \log \ell(\boldsymbol{\theta}^{t+1}; \boldsymbol{y}) \rangle$
16:                Update *domain weights*: $\boldsymbol{\alpha}^{t+1} \leftarrow \dfrac{\hat{\boldsymbol{\alpha}}}{\sum_{k=1}^K \hat{\boldsymbol{\alpha}}_k}$ with $\hat{\boldsymbol{\alpha}} = \boldsymbol{\alpha}^t \odot \exp\left(\dfrac{\gamma_t}{\mu_{\boldsymbol{\alpha}}} \boldsymbol{a}^t\right)$
17:        **else**        $\boldsymbol{\alpha}^{t+1} \leftarrow \boldsymbol{\alpha}^t$
18:        **end if**
19: **end for**
20: **Return** Model parameters $\boldsymbol{\theta}^T$
---

$\approx 4 \times m$ (model parameters, gradients, and two EMA states). The peak memory usage at reweighting steps stores one additional gradient ($m$), leading to $\approx 25\%$ memory overhead.

**Interpreting the Reweighting Mechanism.** The update rules (Equation 3, Equation 4) reveal an elegant negative feedback regulation at the heart of GRAPE's reweighting mechanism: Task weights $\boldsymbol{z}_n$ increase for tasks where the normalized gradient $\nabla \log \ell(\boldsymbol{\theta}, \boldsymbol{y})$ has low alignment with the overall update direction $\boldsymbol{d}_t$, indicating slow progress under the current regime (Equation 3). Subsequently, domain weights $\boldsymbol{\alpha}_j$ increase for domains whose gradients $\boldsymbol{g}_k$ align well with the normalized gradients of these prioritized, struggling tasks (represented by $\nabla \log \ell(\boldsymbol{\theta}, \boldsymbol{y})$ where $\boldsymbol{y} \sim \text{mix}(\boldsymbol{z})$) (Equation 4). This mechanism continuously directs training focus towards underperforming tasks by upweighting relevant source domains. The full derivation is provided in Appendix B.

## 2.2 Theoretical Properties of GRAPE

The GRAPE algorithm, formulated as a regularized minimax optimization problem solved via multiplicative updates, possesses desirable theoretical properties under standard assumptions. The update rules (Equation 3, Equation 4) are instances of online mirror descent applied to the variables $(\boldsymbol{\alpha}, \boldsymbol{z})$ of the minimax game. Theorem 2.1 suggests that under strong convexity and smoothness assumptions, the algorithm converges towards the set of Pareto optimal solutions at an $\mathcal{O}(1/T)$ rate, where $T$ is the number of training iterations. However, we acknowledge the limitation imposed by the strong convexity assumption. In practical non-convex scenarios of LLM pretraining, convergence guarantees are typically weaker. The proof of Theorem 2.1 is presented in Appendix C.1.

**Theorem 2.1** (Convergence of GRAPE). *Let the loss functions $l_n(\boldsymbol{\theta})$ be $L$-smooth for all $n \in [N]$ and the norm of stochastic gradients be upper-bounded by $\mathcal{G}$. If the learning rate $\gamma_t$ satisfies $\gamma_t \leq \frac{1}{L}$ and the regularization parameters $\mu_{\boldsymbol{\alpha}}, \mu_{\boldsymbol{z}}$ are chosen such that $\mu_{\boldsymbol{\alpha}} > 0$ and $\mu_{\boldsymbol{z}} > 0$, then the GRAPE algorithm with update rules given by the above equations converges to a neighborhood of the Pareto optimal solution at a rate of $\mathcal{O}(1/T)$, where $T$ is the number of training iterations. Specifically, for any $\varepsilon > 0$, there exists $T_0$ such that for all $T > T_0$:*

$$\min_{t \in [T]} \left\{ \max_{n \in [N]} \mathbb{E}[l_n(\boldsymbol{\theta}_t)] - \min_{\boldsymbol{\theta}} \max_{n \in [N]} l_n(\boldsymbol{\theta}) \right\} \leq \varepsilon$$

Furthermore, Theorem 2.2 posits that under similar assumptions of smoothness and bounded gradients, the variance of task performances, $\sigma_t^2 = \mathrm{Var}_{n \in [N]}(l_n(\boldsymbol{\theta}_t))$, tends to be monotonically decrease after an initial phase $\sigma_{t+1}^2 \leq \sigma_t^2$ for $t \geq T_0$. It indicates that the algorithm actively counteracts the divergence of the task performances, which promotes more uniform progress across the task suite compared to static weighting or simple averaging. The proof is presented in Appendix C.2.

**Theorem 2.2** (Monotonic Variance Reduction of Task Performance). *Let $\sigma_t^2 = \mathrm{Var}_{n \in [N]}(l_n(\boldsymbol{\theta}_t))$ denote the variance of task performances at iteration $t$. Let the loss functions $l_n(\boldsymbol{\theta})$ be $L$-smooth for all $n \in [N]$ and the norm of stochastic gradients be upper-bounded by $\mathcal{G}$ , and assuming the task losses are $L$-smooth and $\mu$-strongly convex, the variance decreases monotonically until reaching a minimal basin, i.e., $\sigma_{t+1}^2 \leq \sigma_t^2$ for all $t \geq T_0$ for some finite $T_0$.*

## 3 Experiments

To evaluate the efficacy of GRAPE, we conduct experiments in two distinct multi-target pretraining scenarios: (1) optimizing a generic English language model for diverse reasoning tasks, and (2) optimizing the data mixture from mainstream languages for low-resource language modeling. For all experiments, we train various scales of decoder-only transformers following Vaswani et al. [2023] from scratch. More details on architecture and hyperparameters are provided in Appendix D.1.

### 3.1 Domain Reweighting for Multi-task Reasoning

**Setup.** We consider two pretraining corpora, `ClimbLab` [Diao et al., 2025] with $K$=20 source domains clustered by topics; and `SlimPajama` with $K$=7 domains classified by collection sources. We first apply GRAPE to standard English language model pretraining, where the objective is to dynamically adapt the data mixture to maximize performance across a suite of reasoning benchmarks simultaneously. We select $N$=6 diverse reasoning tasks spanning scientific, logical, physical, and commonsense reasoning: ARC-Easy (ARC-E), ARC-Challenge (ARC-C) [Clark et al., 2018],

SciQ [Welbl et al., 2017], PIQA [Bisk et al., 2019], LogiQA [Liu et al., 2020], and HellaSwag [Zellers et al., 2019]. For each target task $\mathcal{T}_n$, we use its standard validation set to compute the task loss $l_n(\boldsymbol{\theta}_t)$ and the Rate-of-Improvement $r_n^{(t)}$ needed for GRAPE's updates during training. We update task weights $\boldsymbol{z}$ every $\Delta T_{\boldsymbol{z}} = 100$ steps and domain weights $\boldsymbol{\alpha}$ every $\Delta T_{\boldsymbol{\alpha}} = 100$ steps. Initial weights $\boldsymbol{\alpha}^0$ and $\boldsymbol{z}^0$ are set to uniform distributions. We use the AdamW optimizer with standard hyperparameters for LLM pretraining. Experiments were conducted using $4 \times$ H100 80GB GPUs. Implementation details and hyperparameter settings are provided in Appendix D. We report the results on ClimbLab in the following sections and present the results on SlimPajama in subsection D.3.

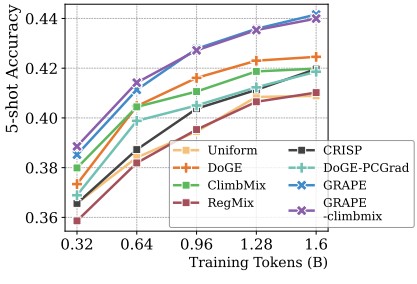

(a) 125M-ClimbLab

**Baselines.** On experiments with $0.7$ B scale models, we compare GRAPE to two baseline methods: (1) **Uniform**: training data are sampled uniformly from each source domains; (2) **DoGE** [Fan et al., 2024a]: domain weights are dynamically adjusted by gradient alignment with uniform task weights across $N$ target tasks. On small-scale (125M) models, we employ a comprehensive suite of baseline approaches: (3) **DoGE-PCGrad**: a multi-task learning extension of DoGE, where the target gradient are dynamically updated with a gradient surgery algorithm PCGrad [Yu et al., 2020]; (4) **RegMix** [Liu et al., 2025]: predict the domain weights using regression towards the optimal average loss across all target tasks; (5) **CRISP** [Grangier et al., 2025]: determine the domain

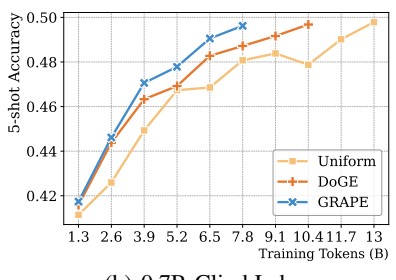

(b) 0.7B-ClimbLab

Figure 1: **GRAPE facilitates multi-task reasoning.** For 125M models, GRAPE and GRAPE-climbmix greatly outperform five baselines; For larger 0.7B models, GRAPE achieves comparable scores as uniform baseline with 40% fewer tokens.

weight distribution by importance sampling, i.e. clustering all data samples from target tasks onto source domains; (6) **ClimbMix** [Diao et al., 2025]: the optimized domain weights curated by iteratively regression and bootstrapping; (7) **GRAPE-CLIMBMIX**: apply GRAPE with domain weights $\boldsymbol{\alpha}$ initialized from ClimbMix weights.

**GRAPE improves multi-task reasoning capability.** According to Table 1 and Figure 1, GRAPE demonstrates consistent improvements on multi-task reasoning capabilities across different model scales, achieving superior average 5-shot reasoning accuracy and learning efficiency compared to various baseline methods. According to Figure 3.1, GRAPE and GRAPE-climbmix, significantly outperform a suite of baselines including DOGE, DOGE-PCGRAD, ClimbMix, RegMix, CRISP, and uniform domain sampling when training small-scale (125M) models. On 0.7B models (Figure 3.1), both DOGE and GRAPE substantially outperform the uniform sampling baseline, demonstrating the general efficacy of adaptive domain reweighting approaches. Notably, GRAPE achieves comparable average 5-shot accuracies to the uniform baseline while utilizing approximately 40% fewer training tokens. Furthermore, compared to DOGE, GRAPE not only exhibits an approximate 25% acceleration in reaching similar accuracy thresholds, but also obtains more well-rounded improvements across all tasks above uniform baseline, as benefit from its distinctive task reweighting mechanism.

Table 1: **5-shot exact match accuracies(%) on target reasoning tasks.** The scores of 125M (resp. 0.7B) models trained on 1.6B (resp. 7.8B) tokens are reported. The best-performed scores are marked as **Bold**, and the second-best scores are Underlined. GRAPE outperforms other baselines on most of target reasoning tasks.

| 125M-ClimbLab | ARC-C | ARC-E | LogiQA | PIQA | SciQ | Hellaswag | Average | # task > uniform base. |
|---|---|---|---|---|---|---|---|---|
| Uniform | 23.12 | 40.74 | 27.96 | 58.98 | 67.50 | 27.02 | 40.89 | - |
| DOGE | 24.57 | 45.33 | 25.35 | 59.52 | 72.20 | 27.77 | 42.45 | 5 |
| REGMIX | 24.66 | 42.85 | 26.11 | 59.30 | 65.90 | 27.30 | 41.02 | 4 |
| CRISP | 22.70 | 44.32 | **29.65** | 57.34 | 70.40 | 27.40 | 41.97 | 4 |
| CLIMBMIX | 24.83 | 42.85 | 25.19 | 60.55 | 70.80 | 27.67 | 41.98 | 5 |
| DOGE-PCGRAD | 24.66 | 42.63 | 27.19 | 58.71 | 70.50 | 27.50 | 41.86 | 4 |
| GRAPE | **26.11** | **47.14** | 27.65 | 61.10 | **74.40** | **28.56** | **44.16** | 5 |
| GRAPE-CLIMBMIX | 26.02 | 46.72 | 27.19 | **62.24** | 73.80 | 28.01 | 43.99 | 5 |
| 0.7B-ClimbLab | ARC-C | ARC-E | LogiQA | PIQA | SciQ | Hellaswag | Average | # task > uniform |
| Uniform | 28.07 | 52.99 | 28.57 | 62.73 | 84.00 | 32.33 | 48.11 | - |
| DOGE | **29.61** | **57.58** | 25.96 | 61.70 | **86.40** | 31.05 | 48.72 | 3 |
| GRAPE | 28.92 | 55.60 | **28.58** | 65.56 | 84.50 | **34.19** | **49.56** | 6 |

**Task weights evolution reflects learning curriculum.** The evolution of task weights ($z_t$) on 0.7B model, as depicted in Figure 2, reveals the dynamic learning curriculum adopted by GRAPE.

In the early stage, the reading comprehension tasks like ARC-E and ARC-C are mostly up-weighted. As training progresses, in the late stage, physical and commonsense reasoning tasks like PIQA and Hellaswag are steadily prioritized, demonstrating the emergence of a skill-wise learning curriculum that moves from foundational to more complex reasoning abilities. Concurrently, tasks like SciQ and LogiQA consistently receive lower weights. This suggest they may either benefit sufficiently

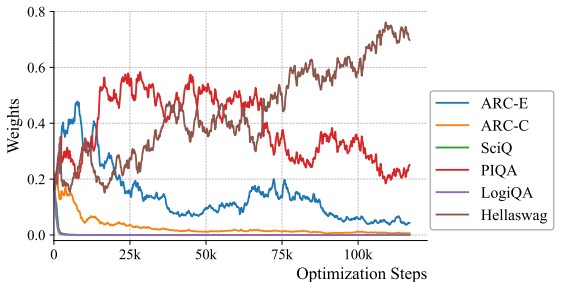

Figure 2: **Task weight evolution of GRAPE.**

from cross-task generalization driven by the prioritized tasks or can be learnt more efficiently relative to other tasks. This adaptive curriculum, by directing resources to evolving bottlenecks, ensures that difficult tasks are not neglected, fostering robust and balanced multi-task reasoning capabilities.

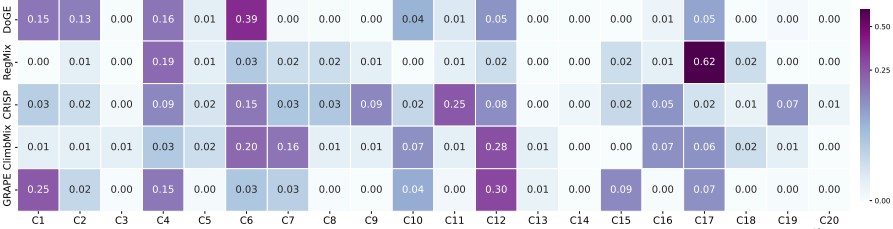

Figure 3: **Domain weights attributions across 20 clusters in the ClimbLab dataset.**

**Which data domains and topics are critical for general reasoning?** Figure 3 presents the domain weight distributions across different methods across 20 data clusters within the ClimbLab corpus, whose main topics are detailed in Table 4. GRAPE notably assigns high importance on Cluster 1,4,10,12, which primarily focus on broad science, mathematics and education contents. It also

explains the constant low-priority on SciQ task indicated in Figure 2 since the training data mixture contains substaintial scientific-QA related contents. In contrast, other methods exhibit different topical biases: DOGE shows a strong preference on Cluster 6, featuring healthcare and genetic contents; REGMIX heavily utilizes Cluster 17 which with topics on health and medical research; CRISP mostly favors Cluster 11, indicating a great emphasis on software and programming. Notably, CRISP achieves significant improvement on LogiQA task, which indicates training on code data is critical to improve the logical reasoning ability of language models.

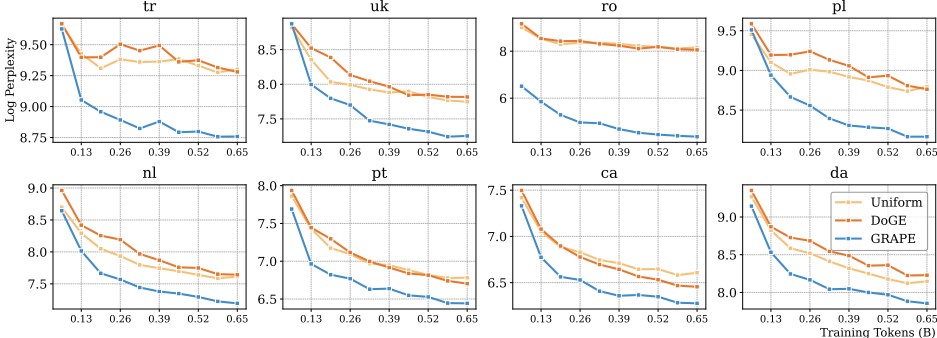

Figure 4: **Low-resource language learning progress by Log-Perplexity.** GRAPE significantly outperforms DOGE and Uniform sampling across all target languages.

## 3.2 Language Mixture for Multi-lingual Learning

**Setup.** In this scenario, we investigate GRAPE's ability to optimize the language composition of the pretraining corpus from mainstream languages to enhance language modeling capabilities across multiple low-resource target languages. The source corpus consists of data from $K = 6$ languages selected from the `wiki40b` dataset [Guo et al., 2020], including high-resource languages English (`en`), French (`fr`), German (`de`), Spanish (`es`), Russian (`ru`) and Italian (`it`). Each language constitutes a domain $\mathcal{D}_i$, and GRAPE adapts the sampling weights $\boldsymbol{\alpha} \in \Delta^k$ over these source languages. We select $N$=8 low-resource languages as target tasks, distinct from the source languages: Turkish (`tr`), Danish (`da`), Catalan (`ca`), Polish (`po`), Romanian (`ro`),

Portuguese (`pt`), Dutch (`nl`), and Ukrian (`uk`). Performance is measured by the language modeling loss, i.e. the log-perplexity (log-PPL), on held-out test sets for each target language. Lower log-PPL indicates better performance.

**GRAPE Enhances Multilingual Adaptation.** Figure 4 illustrates the test log-perplexity on each target low-resource language. GRAPE consistently outperforms both uniform sampling and DOGE baselines across all languages, effectively accelerating the low-resource language modeling. Specifically, GRAPE accelerates the low-resource language modeling by *no less than 60%* in terms of log-perplexity scores, achieving significantly lower final perplexity within the same training budget. While Fan et al. [2024a] showed DOGE greatly improve single-target language learning, its efficacy diminishes when confronted with multiple target languages simultaneously, where it struggles to deliver competitive performance with uniform task weights. This outcome underscores the sub-optimality of average task weighting strate-

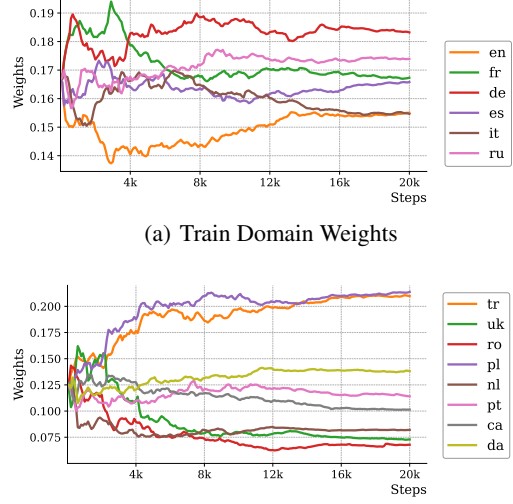

(a) Train Domain Weights

(b) Task Weights

Figure 5: **Weights evolution on multilingual pre-training for low-resource language modeling.**

gies for multi-task learning, especially when tasks exhibit conflicting characteristics. Notably, according to Figure 20, the offline reweighting algorithm such as RegMix and CRISP exhibit a strongly biased performance across various target languages: while RegMix slightly facilitate the learning on most of targets languages including Catalan, Portuguese, Ukrainian and Polish, it sacrifices the performance on Dutch and Danish; the purely embedding-based method CRISP only accelerates the learning on Turkish while sabotaging all the other languages.

**Weight trajectory reveal linguistic relations.** The dynamic interplay between task weights ($z_t$) for target languages (Figure 5, a) and domain weights ($\alpha_t$) for source languages (Figure 5, b) offers valuable insights into GRAPE's adaptive learning strategy and its implicit discovery of inter-lingual relationships. For instance, the initial prioritization of target languages Ukrainian (uk; East Slavic) and Romanian (ro; Romance) corresponds with an increased weighting of linguistically related source languages, namely Russian (ru; East Slavic) and Italian (it; Romance). Subsequently, the sustained high task weight for Polish (pl; Balto-Slavic) aligns with the continued high domain weight of Russian, from the same broad linguistic family. For Turkish (tr), a Turkic language without close phylogenetic relatives within the predominantly Indo-European source domains, its persistent learning challenge is evident from its high task weight. Nevertheless, GRAPE facilitates its learning by strategically up-weighting source languages like German (de) and French (fr), presumably due to their rich linguistic feature sets or similarity in syntax structures.

## 4 Discussion and Limitations

**Progress measurements for Group DRO.** We evaluate the impact of different step-wise progress metrics on the performance of Group DRO within the GRAPE framework. In addition to the primary Rate-of-Improvement (ROI) metric (Equation 1), we investigate two alternatives: *Gap-of-Improvement* (*GOI*) and *EMA-Rate-of-Improvement* (*ROI-ema*). The step-wise improvmenet of task $\mathcal{T}_n$ at step $t$ are assessed as:

$$GOI_n^{(t)} := l_n(\boldsymbol{\theta}_t) - l_n(\boldsymbol{\theta}_{t+1}), \quad ROI\text{-}ema_n^{(t)} := \frac{l_n(\boldsymbol{\theta}_t) - l_n(\boldsymbol{\theta}_{t+1})}{l_{ema,n}^t}, \qquad \forall n \in [N] \quad (5)$$

The exponential moving average loss $l_{ema,n}^t$ is updated as: $l_{ema,n}^t = \beta \cdot l_{ema,n}^{t-1} + (1-\beta) \cdot l_n(\boldsymbol{\theta}_t)$, where the hyperparameter $\beta$ is set to $0.7$ in our experiments. Substituting these alternative progress metrics for $r_n$ in the minimax objective Equation 2 yields modified update rules according to Equation 8 and 9. The core optimization principle for adjusting task weights ($z$) and domain weights ($\alpha$) remains, but the specific gradient terms within the exponents change. More details are provided in Appendix E.1.

Results on the six reasoning tasks (Figure 6) indicate that using *GOI* (GRAPE-gap) prioritizes intrinsically easier tasks. This is attributed to the fact that tasks with larger loss values can exhibit larger absolute improvements ($l_n(\boldsymbol{\theta}_t) - l_n(\boldsymbol{\theta}_{t+1})$) while their relative progress is slow. The *ROI* metric, by normalizing improvement with the current loss $l_i(\boldsymbol{\theta}_t)$, effectively mitigates this bias, offering a fair as-

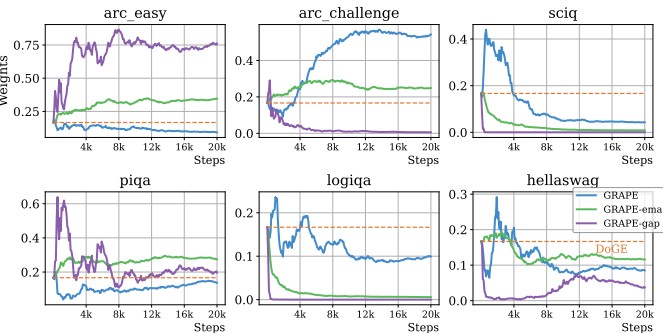

Figure 6: **Task weights evolution on 6 reasoning tasks.**

sessment of task learning progress less biased to easy ones. In comparison, the *ROI*-ema metric, which normalizes the absolute improvement by an exponential moving average of past losses, introduces a historical dependency. While this can smooth out noisy step-wise improvements, our experiments suggest it may also lead to slower adaptation of task weights since the smoothed historical loss might not accurately reflect the current learning state. Comprehensive results across all 12 evaluated task combinations are provided in Appendix E, further substantiating these observations.

**GRAPE outperforms baselines on diverse sets of target tasks.** We further assess the robustness and adaptability of GRAPE on 12 diverse sets of target tasks with varying degrees of diversity and complexity. The task configurations and detailed results are presented in Appendix E. According to Figure 27, to support both code comprehension (Kodcode) and mathematical reasoning (GSM8K) alongside commonsense reasoning task (Hellaswag), GRAPE effectively balance code-specific sources (e.g. GitHub, Stackexchange) and technical web text (e.g. CommonCrawl and C4) for mathematical reasoning with broader corpora. With high task diversity on 8 tasks Figure 28 where commonsense reasoning tasks are predominant, GRAPE largely upweights of general knowledge sources like the Book domain to cater to the majority need, while maintaining a smaller yet significant allocation from domain-specific sources (e.g., GitHub, StackExchange). The comprehensive results in Table 2 and Appendix E demonstrate that GRAPE maintains strong average performance on

diverse task combinations with various complexities, outperforming baselines that struggle to cater to disparate learning objectives with fixed data mixtures and uniform task priorities.

Table 2: **Average test log-perplexity (↓) on 12 task combinations.** GRAPE and the variants outperform uniform and DoGE across all task sets; on 7 out of 12 tasks, GRAPE outperforms other two DRO variants.

| Method | T1 | T2 | T3 | T4 | T5 | T6 | T7 | T8 | T9 | T10 | T11 | T12 |
|---|---|---|---|---|---|---|---|---|---|---|---|---|
| Uniform | 4.30 | 4.01 | 4.11 | 3.75 | 3.85 | 3.62 | 3.20 | 3.90 | 4.33 | 4.62 | 4.81 | 4.53 |
| DoGE | 4.18 | 4.94 | 4.00 | 3.66 | 3.70 | 3.63 | 3.12 | 3.83 | 4.13 | 4.44 | 4.57 | 4.33 |
| GRAPE | **3.53** | **3.61** | 3.70 | 3.39 | **3.36** | **3.03** | **2.93** | **3.45** | **3.85** | 3.94 | 3.93 | 3.70 |
| GRAPE-ema | 3.68 | 3.69 | 3.73 | **3.37** | 3.37 | 3.09 | 2.96 | 3.53 | 3.88 | **3.91** | 3.90 | 3.73 |
| GRAPE-gap | 3.58 | 3.68 | **3.68** | 3.42 | **3.36** | 3.05 | 3.11 | 3.51 | 3.88 | 3.86 | **3.85** | **3.59** |

**Hyperparameter tuning.** The performance and efficiency of GRAPE is dependent on its hyper-parameters, including the regularization coefficients $\mu_\alpha$ and $\mu_z$, as well as the update frequencies $\Delta T_\alpha$ and $\Delta T_z$. Due to the high resource demands associated with LLM pretraining, an exhaustive hyperparameter sweep was beyond the scope of the current work. We posit that the performance of GRAPE can be further improved through extensive hyperparameter tuning, tailored to specific datasets and task configurations. How to determine a theoretical near-optimal step-sizes of domain weights and task weights updates is a direction for future work.

**Impact of training data quality and domain granularity.** Our investigations reveal that the quality of the pretraining corpus and the granularity of its domain partitions critically influence the efficacy of domain reweighting algorithms, including GRAPE, DoGE, and offline methods such as REGMIX and CRISP. On ClimbLab with higher-quality data and fine-grained semantic-based clustering, the domain weights exhibit more precise and impactful adjustments in response to the task curriculum compared to the coarser, source-defined domains of SlimPajama, where the benefits of reweighting can be less distinct. Notably, the offline reweighting algorithm such as RegMix and CRISP are prone to over-concentrated on some particular domains, which leads to biased performance in multi-task learning. These findings underscore the importance of meticulously partitioned data domains for the benefits of adaptive data mixing strategies, suggesting a potential direction for future research.

**Towards better understandings of scaling effects on learning curriculum.** The efficacy of domain reweighting are observed to vary significantly with model scale, besides the domain granularity of the training corpus. The 0.7B model tended to develop a highly concentrated focus on a single challenging task in later training stages, whereas the 125M model maintained a more distributed prioritization across several tasks (10). This suggests that the larger-scale model might be more sensitive to the conflicts between targets. Understanding these scaling effects, how the ideal curriculum changes with model size, and how its efficacy is bounded by data quality and partitioning, is crucial to developing more robust and universally effective adaptive pretraining strategies.

**Towards a sample-level DRO framework for finer-grained reasoning.** The current formulation of GRAPE applies Group Distributed Robust Optimization (DRO) at the task level, prioritizing data from entire target tasks that exhibit slower learning progress. A promising avenue for future research is the extension of this framework to sample-level DRO, where individual data instances (e.g., specific questions or examples) would be dynamically identified and assigned higher importance. By directing attention to the *hardest* samples, it allows the model to focus intensely on correcting specific errors in the mid- or post-training stages, or during interactions with evolving environment.

## 5   Conclusion

This paper introduced GRAPE, a novel multi-target domain reweighting algorithm designed to calibrate pretraining data mixture for robust performance across diverse target tasks. By dynamically adjusting domain and task weights through a Group DRO minimax framework, GRAPE effectively prioritizes tasks demonstrating the slowest improvement. Empirical results on challenging reasoning tasks and multilingual modeling showcase GRAPE's significant advantages over existing baselines, highlighting the efficacy of its adaptive curriculum for multi-target learning.

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

# Contents

# A  Related Work

**Data Reweighting for LLM Pretraining**  Pretraining aims to create general-purpose models, requiring training on massive datasets. Data selection methods can be used to determine the optimal dataset composition according to various objective functions. They operate at different granularities with domain-level data selection methods allowing for less fine-grained control than token-level and sample-level methods but offering more tractable optimization that scales better to the massive datasets typical for LLMs. Learned approaches for domain reweighting either determine fixed domain weights [Xie et al., 2023, Fan et al., 2024a] or dynamically adjust the weights during training of the final model [Fan et al., 2024b, Chen et al., 2023, Wang et al., 2020, Albalak et al., 2023]. Fixed-weight approaches rely on proxy models which require significant upfront compute and assume transferability of domain weights across model and training data scales. Notably, RegMix [Liu et al., 2025] trains hundreds of proxy models under tight computational budgets to fit a regressor predicting the optimal mixture for a held-out validation domain, then employs the top-ranked mixture during large-scale training; it relies on a single static surrogate objective (Pile-CC validation loss) that may not generalize to varied downstream tasks. CLIMB [Diao et al., 2025] similarly uses regression but first partitions the entire pretraining corpus into 20 semantically coherent "domains" via unsupervised clustering of document embeddings and then runs a more efficient iterative mixture search to optimize their weights. DoGE [Fan et al., 2024a] uses a small proxy model to learn static domain weights based on bilevel optimization and gradient alignment, upweighting domains whose gradients most closely align with the chosen target domain. DGA [Fan et al., 2024b] extends this approach to the online scenario by periodically updating weights based on gradient alignment on the large model.

Domain reweighting is in general a two-step process, where we first define an objective that captures model performance, and secondly optimize domain weights according to that objective. Existing methods consider only single-objective scenarios that optimize validation loss of a single target as a proxy for downstream task performance. They lack mechanisms to balance improvements across multiple target tasks with competing objectives. Our approach provides a method to balance multiple tasks during pretraining, building on the gradient alignment score idea and using validation loss of downstream tasks as a surrogate to downstream task performance.

**Gradient Surgery for Multi-task Learning.**  The multi-task learning (MTL) problem has been intensively explored in literature [Yu et al., 2020, Liu et al., 2024, Chen et al., 2018, Navon et al., 2022], while most of the proposed methods requires sufficient amount of training data directly from the target tasks. Given $N \geq 2$ different tasks associated with loss function $l_i(\boldsymbol{\theta})$, the goal is to optimize model parameters $\boldsymbol{\theta}$ that perform well across all target tasks. A standard objective for MTL is minimizing the average loss over all tasks. However, optimizing this averaged loss objective often lead to suboptimal and biased multi-tasking performance because of tasks with conflicting gradients or dominant gradients with large magnitudes [Yu et al., 2020, Liu et al., 2024]. Existing methods aim to avoid gradient conflict by gradient surgery, where the target gradients are linearly combined with adaptive weights Chen et al. [2018], Yu et al. [2020]. For instance, PCGrad [Yu et al., 2020] heuristically removes inter-task gradient conflicts via projections at cost $\mathcal{O}\left(|\boldsymbol{\theta}| N^2\right)$. CAGrad [Liu et al., 2024] strikes a balance between worst-task improvement and average loss reduction by solving a constrained optimization problem that requires $\mathcal{O}(N^3)$ operations. MGDA [Désidéri, 2012] frames MTL as a multi-objective optimization and solves a quadratic programming problem, guaranteeing Pareto-optimal updates. NASH-MTL [Navon et al., 2022] offers equivalent Pareto guarantees, by treating gradient conflicts as a bargaining game that maximizes the product of task improvements (sum of log utility functions). All these methods share a common drawback : substantial memory requirements to store task gradients, $O\left(|\boldsymbol{\theta}| N\right)$, and at least $O\left(|\boldsymbol{\theta}| N^2\right)$ for the optimization step. These computational and memory costs become prohibitive for large-scale models and numerous tasks. FAMO [Liu et al., 2023] attempts to address these efficiency concerns. It dynamically weights task losses using $\mathcal{O}(1)$ space and time per iteration by trying to enforce equal loss decrease rate among tasks. We apply this idea in the domain reweighting setting to develop a multi-task-adaptive domain reweighting method.

# B  Derivation of GRAPE

We start with the minimax problem from Equation (2):

$$\max_{\boldsymbol{\alpha} \in \Delta^k} \min_{\mathbf{z} \in \Delta^N} \gamma_t \sum_{j=1}^{k} \alpha_j \sum_{i=1}^{N} z_i \langle \nabla_{\boldsymbol{\theta}} \log l_i(\boldsymbol{\theta}_t), \mathbf{g}_j(\boldsymbol{\theta}_t) \rangle - h(\boldsymbol{\alpha}) + r(\mathbf{z})$$

With the regularization terms defined as:

$$h(\boldsymbol{\alpha}) := \mu_{\boldsymbol{\alpha}} D_{\Psi}(\boldsymbol{\alpha} \| \boldsymbol{\alpha}^{(t-1)})$$
$$r(\mathbf{z}) := \mu_{\mathbf{z}} D_{\Psi}(\mathbf{z} \| \mathbf{z}^{(t-1)})$$

Where $D_{\Psi}$ is the Bregman divergence with $\Psi(\mathbf{b}) = \sum_i b_i \log(b_i)$:

$$D_{\Psi}(\mathbf{a} \| \mathbf{b}) = \sum_i a_i \log \frac{a_i}{b_i} - \sum_i a_i + \sum_i b_i = \sum_i a_i \log \frac{a_i}{b_i}$$

where the last simplification follows because $\sum_i a_i = \sum_i b_i = 1$ for distributions.

For the inner minimization over $\mathbf{z}$, we form the Lagrangian:

$$\mathcal{L}(\mathbf{z}, \lambda) = \gamma_t \sum_{j=1}^{k} \alpha_j^{(t)} \sum_{i=1}^{N} z_i \langle \nabla_{\boldsymbol{\theta}} \log l_i(\boldsymbol{\theta}_t), \mathbf{g}_j(\boldsymbol{\theta}_t) \rangle + \mu_{\mathbf{z}} \sum_{i=1}^{N} z_i \log \frac{z_i}{z_i^{(t-1)}} + \lambda \left( \sum_{i=1}^{N} z_i - 1 \right)$$

Taking the partial derivative with respect to $z_i$ and setting to zero:

$$\frac{\partial \mathcal{L}}{\partial z_i} = \gamma_t \sum_{j=1}^{k} \alpha_j^{(t)} \langle \nabla_{\boldsymbol{\theta}} \log l_i(\boldsymbol{\theta}_t), \mathbf{g}_j(\boldsymbol{\theta}_t) \rangle + \mu_{\mathbf{z}} \left( 1 + \log \frac{z_i}{z_i^{(t-1)}} \right) + \lambda = 0$$

Solving for $z_i$:

$$z_i = z_i^{(t-1)} \exp \left( -\frac{\gamma_t}{\mu_{\mathbf{z}}} \sum_{j=1}^{k} \alpha_j^{(t)} \langle \nabla_{\boldsymbol{\theta}} \log l_i(\boldsymbol{\theta}_t), \mathbf{g}_j(\boldsymbol{\theta}_t) \rangle - \frac{\lambda + \mu_{\mathbf{z}}}{\mu_{\mathbf{z}}} \right)$$

Let $\mathbf{d}_t = \sum_{j=1}^{k} \alpha_j^{(t)} \mathbf{g}_j(\boldsymbol{\theta}_t)$ be the update direction. Then:

$$z_i = z_i^{(t-1)} \exp \left( -\frac{\gamma_t}{\mu_{\mathbf{z}}} \langle \nabla_{\boldsymbol{\theta}} \log l_i(\boldsymbol{\theta}_t), \mathbf{d}_t \rangle - \frac{\lambda + \mu_{\mathbf{z}}}{\mu_{\mathbf{z}}} \right)$$

Using the constraint $\sum_{i=1}^{N} z_i = 1$ to solve for $\lambda$:

$$\sum_{i=1}^{N} z_i^{(t-1)} \exp \left( -\frac{\gamma_t}{\mu_{\mathbf{z}}} \langle \nabla_{\boldsymbol{\theta}} \log l_i(\boldsymbol{\theta}_t), \mathbf{d}_t \rangle - \frac{\lambda + \mu_{\mathbf{z}}}{\mu_{\mathbf{z}}} \right) = 1$$

Let $Z_{\mathbf{z}} = \sum_{i=1}^{N} z_i^{(t-1)} \exp \left( -\frac{\gamma_t}{\mu_{\mathbf{z}}} \langle \nabla_{\boldsymbol{\theta}} \log l_i(\boldsymbol{\theta}_t), \mathbf{d}_t \rangle \right)$, then $\exp \left( -\frac{\lambda + \mu_{\mathbf{z}}}{\mu_{\mathbf{z}}} \right) = \frac{1}{Z_{\mathbf{z}}}$.

Substituting back, we get the update rule for task weights:

$$z_i^{(t)} = \frac{z_i^{(t-1)} \exp \left( -\frac{\gamma_t}{\mu_{\mathbf{z}}} \langle \nabla_{\boldsymbol{\theta}} \log l_i(\boldsymbol{\theta}_t), \mathbf{d}_t \rangle \right)}{Z_{\mathbf{z}}}$$

Now, for the outer maximization over $\boldsymbol{\alpha}$, we substitute the optimal $\mathbf{z}^{(t)}$ back into the original objective function. Let $f_{ij} = \langle \nabla_{\boldsymbol{\theta}} \log l_i(\boldsymbol{\theta}_t), \mathbf{g}_j(\boldsymbol{\theta}_t) \rangle$.

The resulting optimization problem becomes:

$$\max_{\boldsymbol{\alpha} \in \Delta^k} \gamma_t \sum_{j=1}^{k} \alpha_j \sum_{i=1}^{N} z_i^{(t)} f_{ij} - \mu_{\boldsymbol{\alpha}} \sum_{j=1}^{k} \alpha_j \log \frac{\alpha_j}{\alpha_j^{(t-1)}}$$

We form the Lagrangian for this maximization problem:

$$\mathcal{L}(\boldsymbol{\alpha}, \nu) = \gamma_t \sum_{j=1}^{k} \alpha_j \sum_{i=1}^{N} z_i^{(t)} f_{ij} - \mu_{\boldsymbol{\alpha}} \sum_{j=1}^{k} \alpha_j \log \frac{\alpha_j}{\alpha_j^{(t-1)}} + \nu \left( \sum_{j=1}^{k} \alpha_j - 1 \right)$$

Taking the partial derivative with respect to $\alpha_j$ and setting to zero:

$$\frac{\partial \mathcal{L}}{\partial \alpha_j} = \gamma_t \sum_{i=1}^{N} z_i^{(t)} f_{ij} - \mu_{\boldsymbol{\alpha}} \left( 1 + \log \frac{\alpha_j}{\alpha_j^{(t-1)}} \right) + \nu = 0$$

Solving for $\alpha_j$:

$$\alpha_j = \alpha_j^{(t-1)} \exp \left( \frac{\gamma_t}{\mu_{\boldsymbol{\alpha}}} \sum_{i=1}^{N} z_i^{(t)} f_{ij} - \frac{\nu + \mu_{\boldsymbol{\alpha}}}{\mu_{\boldsymbol{\alpha}}} \right)$$

Using the constraint $\sum_{j=1}^{k} \alpha_j = 1$ to solve for $\nu$:

$$\sum_{j=1}^{k} \alpha_j^{(t-1)} \exp \left( \frac{\gamma_t}{\mu_{\boldsymbol{\alpha}}} \sum_{i=1}^{N} z_i^{(t)} f_{ij} - \frac{\nu + \mu_{\boldsymbol{\alpha}}}{\mu_{\boldsymbol{\alpha}}} \right) = 1$$

Let $Z_{\boldsymbol{\alpha}} = \sum_{j=1}^{k} \alpha_j^{(t-1)} \exp \left( \frac{\gamma_t}{\mu_{\boldsymbol{\alpha}}} \sum_{i=1}^{N} z_i^{(t)} f_{ij} \right)$, then $\exp \left( -\frac{\nu + \mu_{\boldsymbol{\alpha}}}{\mu_{\boldsymbol{\alpha}}} \right) = \frac{1}{Z_{\boldsymbol{\alpha}}}$.

Substituting back, we get the update rule for gradient weights:

$$\alpha_j^{(t)} = \frac{\alpha_j^{(t-1)} \exp \left( \frac{\gamma_t}{\mu_{\boldsymbol{\alpha}}} \sum_{i=1}^{N} z_i^{(t)} f_{ij} \right)}{Z_{\boldsymbol{\alpha}}}$$

Expanding $f_{ij}$ back to its full form:

$$\alpha_j^{(t)} = \frac{\alpha_j^{(t-1)} \exp \left( \frac{\gamma_t}{\mu_{\boldsymbol{\alpha}}} \sum_{i=1}^{N} z_i^{(t)} \langle \nabla_{\boldsymbol{\theta}} \log l_i(\boldsymbol{\theta}_t), \mathbf{g}_j(\boldsymbol{\theta}_t) \rangle \right)}{Z_{\boldsymbol{\alpha}}}$$

# C  Theoretic Properties of GRAPE

## C.1  Proof of the Convergence Theorem

**Theorem C.1** (Convergence of GRAPE). *Let the loss functions $l_n(\boldsymbol{\theta})$ be L-smooth for all $n \in [N]$ and the norm of stochastic gradients be upper-bounded by $\mathcal{G}$. If the learning rate $\gamma_t$ satisfies $\gamma_t \leq \frac{1}{L}$ and the regularization parameters $\mu_{\boldsymbol{\alpha}}$, $\mu_{\boldsymbol{z}}$ are chosen such that $\mu_{\boldsymbol{\alpha}} > 0$ and $\mu_{\boldsymbol{z}} > 0$, then the GRAPE algorithm with update rules given by the above equations converges to a neighborhood of the Pareto optimal solution at a rate of $\mathcal{O}(1/T)$, where $T$ is the number of training iterations. Specifically, for any $\varepsilon > 0$, there exists $T_0$ such that for all $T > T_0$:*

$$\min_{t \in [T]} \left\{ \max_{n \in [N]} \mathbb{E}[l_n(\boldsymbol{\theta}_t)] - \min_{\boldsymbol{\theta}} \max_{n \in [N]} l_n(\boldsymbol{\theta}) \right\} \leq \varepsilon$$

*Proof.* We establish the convergence of GRAPE by analyzing the dynamics of the minimax optimization problem and demonstrating that the algorithm makes consistent progress toward Pareto optimal solutions while balancing performance across tasks. First, we analyze the task weight updates. Given that $\boldsymbol{z}$ is updated according to:

$$z_n^t = z_n^{t-1} \cdot \exp\left( -\frac{\gamma_t}{\mu_{\boldsymbol{z}}} \langle \boldsymbol{d}_t, \nabla_{\boldsymbol{\theta}} \log l_n(\boldsymbol{\theta}_t) \rangle \right) \tag{6}$$

The task weights adjust to increase focus on tasks with slower improvement rates. For any task $i$ where $\langle \boldsymbol{d}_t, \nabla_{\boldsymbol{\theta}} \log l_n(\boldsymbol{\theta}_t) \rangle < 0$ (indicating poor alignment between the current update direction and task gradient), $z_n^t$ increases proportionally. This ensures more attention to tasks making less progress.

For each iteration $t$, let $\mathcal{L}(\boldsymbol{\theta}_t) = \max_{n \in [N]} l_n(\boldsymbol{\theta}_t)$ represent our worst-case objective. Let $n_t = \arg\max_{n \in [N]} l_n(\boldsymbol{\theta}_t)$ be the index of the task with highest loss at iteration $t$. Due to the exponential update rule, as $t$ increases, $z_{n_t}^t$ approaches 1, while other weights approach 0, directing optimization toward the current worst-case task. Considering the domain weight updates:

$$\alpha_k^t = \alpha_k^{t-1} \cdot \exp\left( \gamma_t \sum_{n=1}^{N} \frac{z_n^{t-1}}{\mu_{\boldsymbol{\alpha}}} \langle \boldsymbol{g}_k(\boldsymbol{\theta}_t), \nabla_{\boldsymbol{\theta}} \log l_n(\boldsymbol{\theta}_t) \rangle \right)$$

These updates increase weights for domains whose gradients align well with the task gradients weighted by $\boldsymbol{z}$. Since $\boldsymbol{z}$ concentrates on the worst-performing tasks, $\boldsymbol{\alpha}$ increasingly favors domains beneficial to these tasks. Given the $L$-smoothness of the loss functions, we can upper-bound the progress on the worst-case objective:

$$\mathcal{L}(\boldsymbol{\theta}_{t+1}) - \mathcal{L}(\boldsymbol{\theta}_t) \leq \langle \nabla_{\boldsymbol{\theta}} l_{n_t}(\boldsymbol{\theta}_t), -\gamma_t \boldsymbol{d}_t \rangle + \frac{L\gamma_t^2}{2} \|\boldsymbol{d}_t\|^2$$

Let $\mathcal{Q}_t = \langle \nabla_{\boldsymbol{\theta}} l_{n_t}(\boldsymbol{\theta}_t), \boldsymbol{d}_t \rangle$. Due to our update rules for $\boldsymbol{z}$ and $\boldsymbol{\alpha}$, as $t$ increases, $\mathcal{Q}_t$ becomes increasingly positive (as domain weights shift toward domains beneficial for the worst-performing task). With the condition $\gamma_t \leq \frac{1}{L}$, we have:

$$\mathcal{L}(\boldsymbol{\theta}_{t+1}) - \mathcal{L}(\boldsymbol{\theta}_t) \leq -\gamma_t \mathcal{Q}_t + \frac{\gamma_t}{2} \|\boldsymbol{d}_t\|^2$$

Since the domain weights are probability distributions and gradients are bounded (due to $L$-smoothness), $\|\boldsymbol{d}_t\|$ is bounded by some constant $G$. Therefore:

$$\mathcal{L}(\boldsymbol{\theta}_{t+1}) - \mathcal{L}(\boldsymbol{\theta}_t) \leq -\gamma_t \mathcal{Q}_t + \frac{\gamma_t G^2}{2}$$

As optimization progresses and $\mathcal{Q}_t$ increases due to our reweighting strategy, we eventually reach a point where $\mathcal{Q}_t > \frac{G^2}{2}$, ensuring consistent progress on the worst-case objective. Summing over iterations and applying the $\mu$-strong convexity, we obtain:

$$\sum_{t=1}^{T} (\mathcal{L}(\boldsymbol{\theta}_{t+1}) - \mathcal{L}(\boldsymbol{\theta}_t)) \leq \sum_{t=1}^{T} \left( -\gamma_t \mathcal{Q}_t + \frac{\gamma_t G^2}{2} \right)$$

This implies:

$$\mathcal{L}(\boldsymbol{\theta}_{T+1}) - \mathcal{L}(\boldsymbol{\theta}_1) \leq \sum_{t=1}^{T} \gamma_t \left( -\mathcal{Q}_t + \frac{G^2}{2} \right)$$

Given our reweighting strategy ensures $\mathcal{Q}_t > \frac{G^2}{2}$ after sufficient iterations, this difference becomes negative, establishing convergence.

For the convergence rate, with constant step size $\gamma_t = \frac{1}{L}$, we can show:

$$\min_{t \in [T]} \{\mathcal{L}(\boldsymbol{\theta}_t) - \mathcal{L}^*\} \leq \frac{L\|\boldsymbol{\theta}_1 - \boldsymbol{\theta}^*\|^2}{2T} + \frac{G^2}{2L} - \frac{1}{T} \sum_{t=1}^{T} \frac{\mathcal{Q}_t}{L}$$

Where $\mathcal{L}^* = \min_{\boldsymbol{\theta}} \max_{n \in [N]} l_n(\boldsymbol{\theta})$ and $\boldsymbol{\theta}^*$ is the corresponding optimal parameter. As $T \to \infty$, this bound approaches $\frac{G^2}{2L} - \frac{\bar{\mathcal{Q}}}{L}$, where $\bar{\mathcal{Q}}$ is the average alignment. Therefore, GRAPE converges to a neighborhood of the Pareto optimal solution at a rate of $\mathcal{O}(1/T)$ □

## C.2 Proof of the Variance Reduction Theorem

**Theorem C.2** (Monotonic Variance Reduction of Task Performance). *Let $\sigma_t^2 = \text{Var}_{n \in [N]}(l_n(\boldsymbol{\theta}_t))$ denote the variance of task performances at iteration t. Let the loss functions $l_n(\boldsymbol{\theta})$ be L-smooth for all $n \in [N]$ and the norm of stochastic gradients be upper-bounded by $\mathcal{G}$ , and assuming the task losses are L-smooth and $\mu$-strongly convex, the variance decreases monotonically until reaching a minimal basin, i.e., $\sigma_{t+1}^2 \leq \sigma_t^2$ for all $t \geq T_0$ for some finite $T_0$.*

*Proof.* According to the definition of Rate-of-Improvement (RoI) for task $i$ at iteration $t$:

$$r_i^{(t)} = \frac{l_i(\boldsymbol{\theta}_t) - l_i(\boldsymbol{\theta}_{t+1})}{l_i(\boldsymbol{\theta}_t)}$$

For any task $i$, by the $L$-smoothness assumption, we can bound the change in loss:

$$l_i(\boldsymbol{\theta}_{t+1}) - l_i(\boldsymbol{\theta}_t) \leq -\gamma_t \langle \nabla_{\boldsymbol{\theta}} l_i(\boldsymbol{\theta}_t), \boldsymbol{d}_t \rangle + \frac{L\gamma_t^2}{2} |\boldsymbol{d}_t|^2$$

Let $\Delta l_i(t) = l_i(\boldsymbol{\theta}_{t+1}) - l_i(\boldsymbol{\theta}_t)$ represent the change in loss for task $i$.

First, we establish a direct relationship between task weights and improvement rates. At each iteration, GRAPE updates task weights according to:

$$z_i^t \propto z_i^{t-1} \cdot \exp\left( -\frac{\gamma_t}{\mu_{\boldsymbol{z}}} \langle \boldsymbol{d}_t, \nabla_{\boldsymbol{\theta}} \log l_i(\boldsymbol{\theta}_t) \rangle \right)$$

it reflects the principle that tasks with lower RoI receive higher weights in the next iteration.

When domain weights are subsequently updated, domains that yield better improvements for these higher-weighted (struggling) tasks are upweighted:

$$\alpha_j^t \propto \alpha_j^{t-1} \cdot \exp\left( \gamma_t \sum_{i=1}^{N} \frac{z_i^{t-1}}{\mu_{\boldsymbol{\alpha}}} \langle \boldsymbol{g}_j(\boldsymbol{\theta}_t), \nabla_{\boldsymbol{\theta}} \log l_i(\boldsymbol{\theta}_t) \rangle \right)$$

This creates a feedback mechanism that specifically targets and improves tasks with lower RoI values. This indicates that tasks with lower RoI experience proportionally greater increases in their improvement rates. We can formalize this in **Lemma 1**:

**Lemma 1**: There exists a constant $\beta > 0$ such that if $r_i^{(t)} < r_j^{(t)}$ for tasks $i$ and $j$, then after GRAPE's reweighting mechanism, the expected improvement in the next iteration satisfies: $\mathbb{E}[r_i^{(t+1)}] - \mathbb{E}[r_i^{(t)}] > \mathbb{E}[r_j^{(t+1)}] - \mathbb{E}[r_j^{(t)}] + \beta(r_j^{(t)} - r_i^{(t)})$

We further define the normalized task losses:

$$\hat{l}_i(t) = \frac{l_i(\boldsymbol{\theta}_t) - \min_j l_j(\boldsymbol{\theta}_t)}{\max_j l_j(\boldsymbol{\theta}_t) - \min_j l_j(\boldsymbol{\theta}_t)}$$

We show that the variance of normalized loss $\hat{l}_i(t)$ is proportional to the variance of losses:

**Lemma 2 (Variance of Normalized Losses)**: The variance of normalized losses $\hat{\sigma}_t^2 = \mathrm{Var}_{i \in [N]}(\hat{l}_i(t))$ is directly proportional to the variance of original losses $\sigma_t^2 = \mathrm{Var}_{i \in [N]}(l_i(\boldsymbol{\theta}_t))$.

*Proof.* We begin with the definition of normalized losses:

$$\hat{l}_i(t) = \frac{l_i(\boldsymbol{\theta}_t) - \min_n l_n(\boldsymbol{\theta}_t)}{\max_n l_n(\boldsymbol{\theta}_t) - \min_n l_n(\boldsymbol{\theta}_t)}$$

Let's denote $a_t = \min_n l_n(\boldsymbol{\theta}_t)$ and $b_t = \max_n l_n(\boldsymbol{\theta}_t)$ for brevity. The normalized loss can be written as:

$$\hat{l}_i(t) = \frac{l_i(\boldsymbol{\theta}_t) - a_t}{b_t - a_t}$$

This is an affine transformation of the original losses. By the properties of variance for affine transformations, for any random variable $X$ and constants $c$ and $d$, we have:

$$\mathrm{Var}(cX + d) = c^2 \mathrm{Var}(X)$$

In our case, viewing the task losses as samples from a distribution, we have $c = \frac{1}{b_t - a_t}$ and $d = \frac{-a_t}{b_t - a_t}$. Thus, we have,

$$\hat{\sigma}_t^2 = \mathrm{Var}_{i \in [N]}(\hat{l}i(t)) = \mathrm{Var}_{i \in [N]}\left(\frac{l_i(\boldsymbol{\theta}_t) - a_t}{b_t - a_t}\right) = \frac{1}{(b_t - a_t)^2}\mathrm{Var}_{i \in [N]}(l_i(\boldsymbol{\theta}_t)) = \frac{\sigma_t^2}{(b_t - a_t)^2}$$

Therefore:

$$\hat{\sigma}_t^2 = \frac{\sigma_t^2}{(b_t - a_t)^2}$$

Which establishes a direct proportionality between $\hat{\sigma}_t^2$ and $\sigma_t^2$, with the proportionality constant $\frac{1}{(b_t - a_t)^2}$. $\qquad\square$

Given the relationship between task weights and RoI established in **Lemma 1**, tasks with higher normalized losses tend to have lower RoI values and thus receive higher weights in GRAPE. This leads to a correlation between $\hat{l}_i(t)$ and the subsequent change in normalized loss $\Delta\hat{l}_i(t) = \hat{l}_i(t+1) - \hat{l}_i(t)$.

Specifically, for any pair of tasks $i$ and $j$ where $\hat{l}_i(t) > \hat{l}_j(t)$, GRAPE's reweighting mechanism ensures:

$$\mathbb{E}[\Delta\hat{l}_i(t)] < \mathbb{E}[\Delta\hat{l}_j(t)]$$

Therefore, we can express the change in variance:

$$\hat{\sigma}_{t+1}^2 - \hat{\sigma}_t^2 = \frac{1}{N}\sum_{i=1}^{N}(\hat{l}_i(t+1) - \bar{\hat{l}}_{t+1})^2 - \frac{1}{N}\sum_{i=1}^{N}(\hat{l}_i(t) - \bar{\hat{l}}_t)^2$$

$$= \frac{1}{N}\sum_{i=1}^{N}[(\hat{l}_i(t) + \Delta\hat{l}_i(t) - \bar{\hat{l}}_t - \Delta\bar{\hat{l}}_t)^2 - (\hat{l}_i(t) - \bar{\hat{l}}_t)^2]$$

$$= \hat{\sigma}_{t+1}^2 - \hat{\sigma}_t^2 = \frac{1}{N}\sum_{i=1}^{N}[2(\hat{l}_i(t) - \bar{\hat{l}}_t)(\Delta\hat{l}_i(t) - \Delta\bar{\hat{l}}_t) + (\Delta\hat{l}_i(t) - \Delta\bar{\hat{l}}_t)^2]$$

Given that the tasks with higher normalized loss will experience greater improvements on normalized loss under GRAPE, this correlation term is negative:

$$\frac{1}{N}\sum_{i=1}^{N}(\hat{l}_i(t) - \bar{\hat{l}}_t)(\Delta\hat{l}_i(t) - \Delta\bar{\hat{l}}_t) \le -\kappa\hat{\sigma}_t^2$$

for some constant $\kappa > 0$, representing the strength of GRAPE's balancing effect.

The second term, $\frac{1}{N}\sum_{i=1}^{N}(\Delta\hat{l}_i(t) - \Delta\bar{\hat{l}}_t)^2$, represents the variance of the changes in normalized losses, which is bounded by $O(\gamma_t^2)$ under the smoothness assumption. Thus:

$$\hat{\sigma}_{t+1}^2 - \hat{\sigma}_t^2 \le -2\kappa\hat{\sigma}_t^2 + O(\gamma_t^2)$$

For sufficiently small $\gamma_t$, this difference becomes negative, establishing that $\hat{\sigma}_{t+1}^2 < \hat{\sigma}_t^2$. Following **Lemma 2**, it indicates that $\sigma_{t+1}^2 < \sigma_t^2$. $\qquad\square$

# D Domain Reweighting for Multi-task Reasoning

Table 3: Architecture hyperparameters for various model scales used in the paper. All models are vanilla Transformer decoder-only models.

|  | Layers | Attention heads | Embed dim | Hidden dim | Context len. | learning rate ($\gamma$) | $\gamma/\mu_{\alpha}$ | $\gamma/\mu_z$ | batch size |
|---|---|---|---|---|---|---|---|---|---|
| 1M | 2 | 8 | 256 | 512 | 512 | $4 \times 10^{-4}$ | - | - | 64 |
| 125M | 12 | 12 | 768 | 3072 | 512 | $1.5 \times 10^{-4}$ | 1.5 | 10 | 64 |
| 0.7B | 36 | 20 | 1280 | 5120 | 512 | $1.5 \times 10^{-4}$ | 1.5 | 10 | 128 |

## D.1 Implementation Details

On small-scale experiments with 125M models, we employ cosine learning-rate scheduler with maximum $lr = 1.5e^{-4}$. On large-scale runs with 0.7B models, we adopt the WSD scheduler [Hu et al., 2024] where the learning rate remains constant ($lr_{max} = 1.5e^{-4}$) during training while linearly decaying to $lr_{min} = 1.5e^{-5}$ at the last 20% of total iterations. The model architecture and hyperparameter settings are detailed in Table 3. The full implementation of GRAPE is open-sourced in https://github.com/Olivia-fsm/GRAPE_data_mixture_with_multi_target.

**GRAPE.** For GRAPE-specific parameters, we generally set the regularization coefficients $\mu_{\alpha} = 1e^{-4}$ and $\mu_z = 1.5e^{-5}$, corresponding to the step-size of $\gamma/\mu_{\alpha} = 1.5, \gamma/\mu_z = 10$. Both domain weights ($\alpha$) and task weights ($z$) were periodically updated every $\Delta T_{\alpha} = \Delta T_z = 100$ steps. Initial domain weights ($\alpha^0$) and task weights ($z^0$) were set to uniform distribution, unless otherwise specified. Gradients required for the GRAPE weight updates (e.g., $\nabla \log l_i(\theta_t)$ on target tasks, $g_i(\theta_t)$ on source domains) were estimated stochastically using dedicated mini-batches processed within the update step.

**DOGE.** We implement DOGE following the out-of-domain generalization setting in [Fan et al., 2024a]. Domain weights ($\alpha$) are updated every $\Delta T_{\alpha} = 100$ steps during training according to the alignment of gradients between each training data domain and the average gradient across all target tasks, which is equivalent to applying a uniform task weights ($z$) constantly. We apply the same regularization hyperparameters as GRAPE.

**CLIMBMIX.** We set the domain weights following Diao et al. [2025]. Specifically, we utilize the final iteration weights derived by iterative regression and bootstrapping, which is claimed as the optimal mixture in the original paper.

**GRAPE-CLIMBMIX.** While the GRAPE algorithm initializes domain weights ($\alpha^0$) as the uniform distribution, we implement a variant of GRAPE with $\alpha^0$ initialized with the optimized CLIMBMIX weights [Diao et al., 2025]. The other hyperparameters for reweighting and optimization are kept consistent as GRAPE.

**DOGE-PCGRAD.** We tailor the DOGE [Fan et al., 2024a] method for multi-task learning by dynamically tune the target gradient with a gradient surgery method PCGrad [Yu et al., 2020], where the gradients from each target task are combined to avoid gradient conflicts. PCGrad eliminates conflicting gradient components by projecting each task gradient onto the normal plane of any conflicting task gradient. Conflict is indicated by a negative dot product between target gradients. We process gradients iteratively, taking each task gradient in a randomized sequence and adjusting it when conflicts are detected, continuing until all gradients have been properly modified. The random ordering of gradient processing helps prevent systematic bias toward specific tasks. The final target gradient, $g_{PCGrad}$, is calculated as the average of the processed target gradients. At each reweighting step to calculate a conflict-free target gradient, $g_{PCGrad}$. Domain weights ($\alpha$) are updated every $\Delta T_{\alpha} = 100$ steps during training according to the alignment of gradients between each training data domain and $g_{PCGrad}$. We apply the same regularization hyperparameters as GRAPE. With $N$ target tasks, each reweighting step requires storing $N$ gradients into memory, which incures $N \times |\theta|$ storage costs. Thus, the traditional gradient surgery algorithm for multi-task learning can lead to OOM issue due to the significant GPU memory load, which prevent it from scaling up to large models and datasets.

**REGMIX.** We implement REGMIX [Liu et al., 2025] to predict the optimal domain weights ($\alpha$) by fitting a regression model. Following Liu et al. [2025], we train a total of 768 `TinyLlama-1M` proxy models with with various data mixture configurations: 512 of them are used to fit the regression model, while the other 256 of them are saved for evaluating the performance of the regressor. For each experiment run, we record the average validation loss across all target tasks; we then train the LightGBM regressor to predict the input domain weights which achieves the lowest average loss. The other hyperparameters for proxy model training are set to the default values in [Liu et al., 2025].

**CRISP** We compute embedding-based domain weights using CRISP [Grangier et al., 2025]. For each train and target domain we calculate the sequence embeddings using SBERT `MiniLM-L6-v2`. To avoid overfitting, we select 5 representative centroids for each training domains $D_k, k \in [K]$ respectively using k-means. We then obtain the domain weights by applying a nearest-neighbor ($k$=5) classification on each piece of the samples from all target sets, i.e. mapping every individual target embedding to centroids within each training domains. We finally compute the distribution of target samples located in each training domain to get the final domain weights.

### D.2 Results on ClimbLab

#### D.2.1 Description of Dataset

ClimbLab is a dataset introduced by Diao et al. [2025], which is partitioned into 20 clusters according to semantic-based sequence embeddings. The related topic on each cluster (i.e. train domain) are detailed in Table 4.

Table 4: **Main topics of 20 clusters in ClimbLab corpus.**.

| Cluster ID | Topics |
|---|---|
| 1 | Mathematics, Algorithms, Programming, Software Development, Data Analysis |
| 2 | Books, Education, Writing, Literature, AI Ethics, History, Philosophy |
| 3 | Environmental Education, History, Architecture, Engineering, Classical Music |
| 4 | Education, Teaching, Science, Engineering, Psychology, Special Education |
| 5 | International Trade, Business, Economics, AI Consulting, Ethical Decision Making |
| 6 | Genetics, Biotechnology, AI, Robotics, Aging, Healthcare, Industrial Automation |
| 7 | Chemistry, Insects, Taxonomy, Agriculture, Gardening, Veterinary Science |
| 8 | Gaming, Role-Playing, Board Games, Video Games, Strategy, Fantasy, Virtual Reality |
| 9 | Astronomy, Cosmology, Astrophysics, Space Exploration, Urban Planning |
| 10 | Health, Sleep, Clinical Technology, Healthcare, Fitness, Addiction, Early Childhood Education |
| 11 | Software Development, Programming, Web Development, JavaScript, Databases |
| 12 | Technology, Mathematics, Legal Content, Human Rights, Energy Efficiency, Industrial Equipment |
| 13 | Sports, Cricket, Soccer, Tennis, Basketball, Cultural Heritage, Competition |
| 14 | Music, Instrumental Practice, Guitar, Jazz, Singing, Composition, Music Theory |
| 15 | Film, Cinema, Horror, Sci-Fi, Comics, Literature, Criticism, Philosophy |
| 16 | Sustainability, Climate Change, Renewable Energy, Environmental Conservation |
| 17 | Cardiovascular Health, Medical Research, Immunology, Cancer Prevention, Drug Therapy |
| 18 | Technology, Cybersecurity, Social Media, Privacy, Artificial Intelligence, Cloud Computing |
| 19 | Social Media, Digital Communication, Internet Culture, Misinformation, Psychology |
| 20 | Public Safety, Law Enforcement, Political History, Social Justice, Government |

#### D.2.2 Learning Curriculum from Various Reweighting Approaches

**Domain Weights from RegMix.** We present the domain weights from RegMix in Figure 7 and Figure 8, with different sets of target tasks. The RegMix regressor ends up concentrating almost all of its weight to just two clusters. In both settings, Cluster 17 (advanced biomedical and clinical research content) receives the highest weight. Cluster 15 (film and cultural-arts) gets picked up for the general reasoning benchmarks showing that its narrative and conceptual text aligns with the story-and-commonsense inference style those tasks need. Including MedQA/MathQA, increases the weight of Cluster 9 (space science, urban planning), showing that its physics-heavy, formula-rich science content align well with the formal quantitative reasoning that MathQA requires.

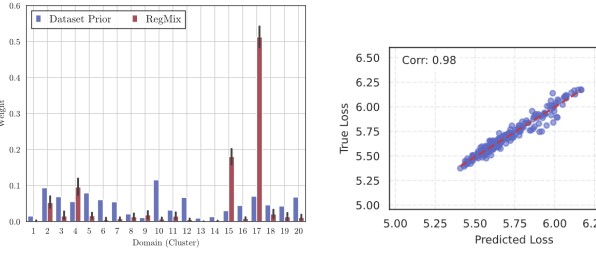

(a) Original vs. RegMix weights    (b) Spearman correlation between predicted v.s. true loss

Figure 7: **RegMix results on ClimbLab with LightGBM regression on 6 reasoning target tasks** [ARC-E, ARC-C, Hellaswag, SciQ, PIQA, LogiQA]

**Domain Weights from CRISP.** We present the domain weights from CRISP in Figure 9, with different sets of target tasks. In both settings CRISP assigns similar weights to the dataset clusters. Main focus is given on code data (Cluster 11), while Cluster 6 with technical life-science and automation content also receives significant attention. Including MedQA and MathQA, slightly

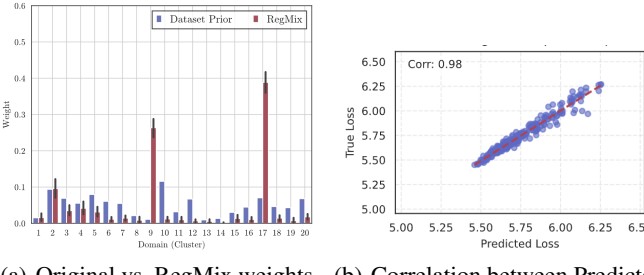

(a) Original vs. RegMix weights  (b) Correlation between Predicted v.s. True loss

Figure 8: **RegMix results on ClimbLab with LightGBM regression on 8 target tasks** [ARC-E, ARC-C, Hellaswag, SciQ, PIQA, LogiQA, MathQA, MedQA].

boosts the weight of Cluster 10 (health and wellness content), reflecting MedQA's embeddings lie closest to clinical-style text.

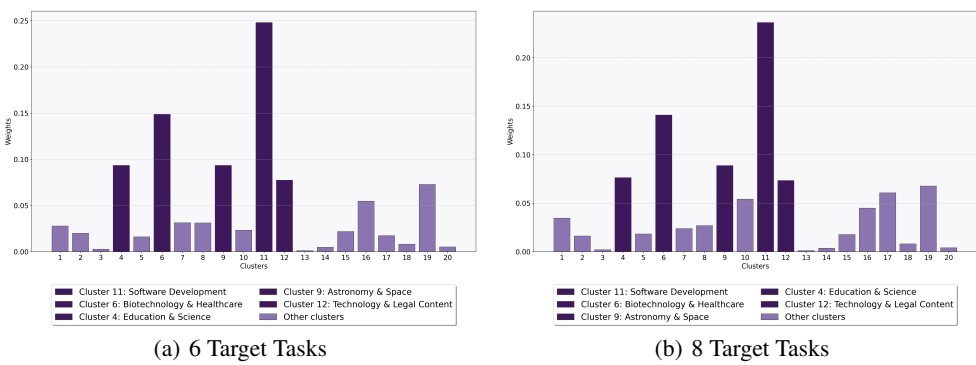

(a) 6 Target Tasks  (b) 8 Target Tasks

Figure 9: CRISP Domain weights across 7 data domains in ClimbLab.

**Domain Weights Evolution from GRAPE and Scaling Effects.** We observed that the task priority differs notably between model sizes. According to Figure 10, the 0.7B parameter model tended to develop a highly concentrated focus on a single challenging task in later training stages, whereas the 125M model maintained a more distributed and stable prioritization across several tasks. This suggests that the larger-scale model might be more sensitive to the conflicts between targets. Additionally, in both cases, the task weight trajectories demonstrate a clear, stage-wise curriculum, which indicates a dynamic, adaptive curriculum is crucial to yield a better multi-task learning performance.

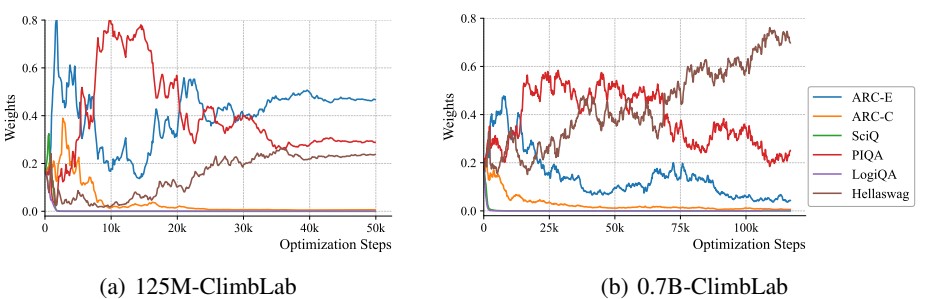

(a) 125M-ClimbLab  (b) 0.7B-ClimbLab

Figure 10: **Task weights trajectory from GRAPE on ClimbLab from** 125M/0.7B **models.**

### D.3 Results on SlimPajama

We present the experimental results on SlimPajama in this section. The hyperparameter and baseline settings are constant with the experiments on ClimbLab.

#### D.3.1 Learning Curriculum from Various Reweighting Approaches

**Domain Weights from REGMIX.** We present the domain weights derived from RegMix in Figure 11 and Figure 12, on various sets of target tasks. In both cases, the RegMix regressor learns a very spiky training mixture, where a large percentage ($\sim 80\%$) are allocated to web-crawled dataset such as C4, while nearly eliminates the domains other than C4, CC, and Book. Compared to 6 common-sense and logical related reasoning tasks, including domain-specific QA tasks - MathQA and MedQA - further increases the weights on C4, which can be attributed to its broad coverage on various knowledge sources.

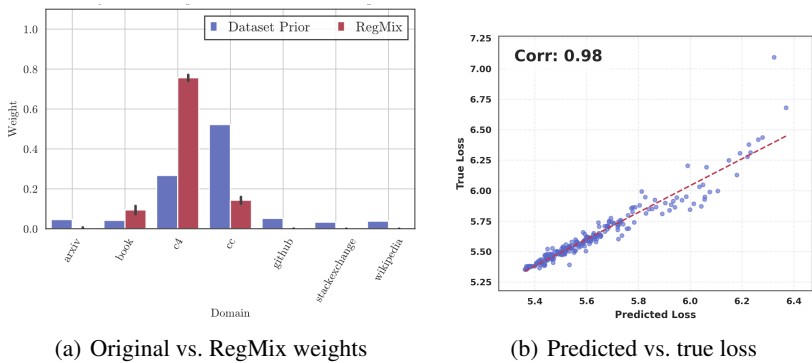

(a) Original vs. RegMix weights       (b) Predicted vs. true loss

Figure 11: **RegMix results on SlimPajama with 6 target tasks** [ARC-E, ARC-C, Hellaswag, SciQ, PIQA, LogiQA]

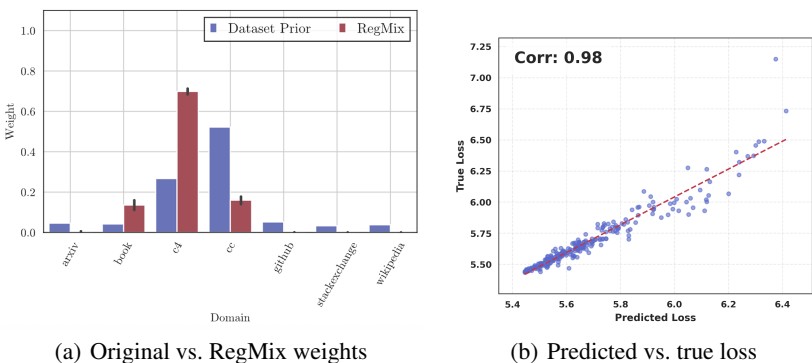

(a) Original vs. RegMix weights       (b) Predicted vs. true loss

Figure 12: **RegMix results on SlimPajama with 8 target tasks** [ARC-E, ARC-C, Hellaswag, SciQ, PIQA, LogiQA, MathQA, MedQA]

**Domain Weights from CRISP.** We present the domain weights derived from CRISP in Figure 13 on various sets of target tasks. CRISP assigns most of the weight to Github, followed by StackExchange. All other domains contribute minimally. A plausible rationale behind the CRISP weight distribution is that short Q&A tasks (ARC-E/C, SciQ, PIQA) embed closest to the concise question-answer format of StackExchange, while multi-step reasoning puzzles (LogiQA) and formula-heavy problems (MathQA) align more with GitHub's structured code snippets, and MedQA's longer clinical passages slightly with Book and arXiv. Including MedQA and MathQA brings clinical and formulaic embeddings that shift a bit of weight back toward Book and arXiv.

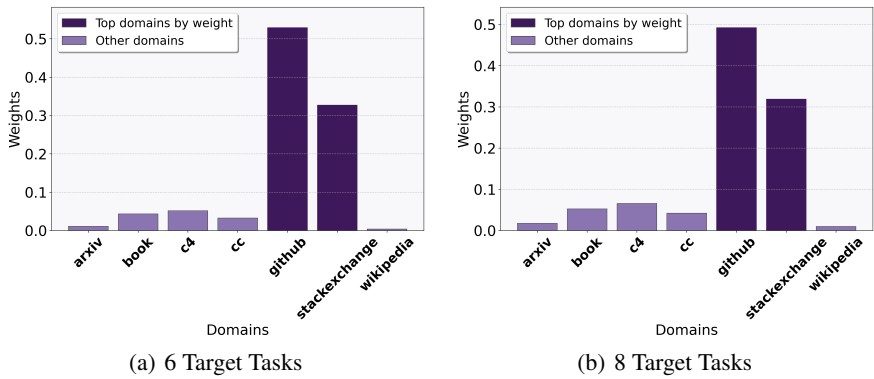

|     (a) 6 Target Tasks     |     (b) 8 Target Tasks     |

Figure 13: CRISP Domain weights across 7 data domains in SlimPajama.

### D.3.2 Results on Various Task Configurations

**Reweighting with 6 Target Tasks: ARC-E, ARC-C, Hellaswag, SciQ, PIQA, LogiQA.** We present the full results on multi-task reasoning experiment, where the data mixture from SlimPajama dataset are adapted towards the optimal performance on 6 target tasks: ARC-E, ARC-C, Hellaswag, SciQ, PIQA, LogiQA. The Log-perplexity and 5-shot accuracy scores on each target reasoning tasks are presented in Figure 14, Table 5 and Table 6, where GRAPE consistently achieves better (lower) perplexity compared to the other 5 baselines. In contrast, CRISP, as an purely embedding-based reweighting algorithm, leads to significantly worse results on all benchmarks, in terms of perplexity and 5-shot accuracy scores.

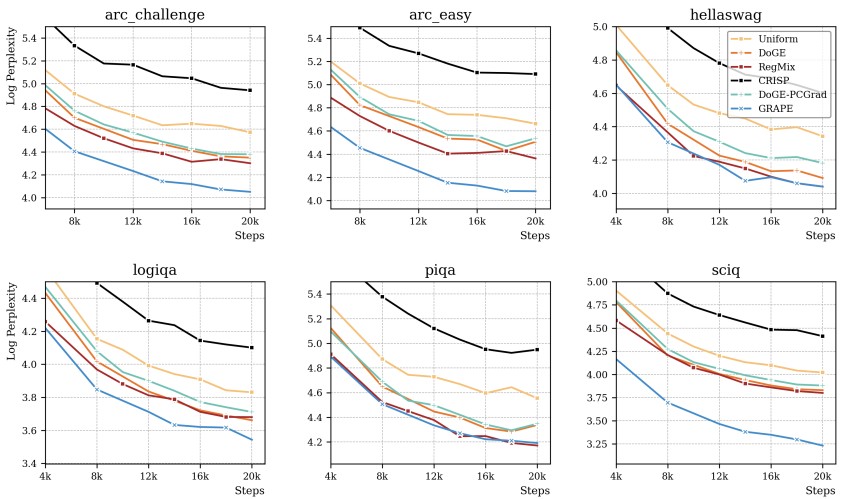

Figure 14: Log-Perplexities on 6 target reasoning tasks

Table 5: **Log perplexity scores** (↓). Models (125M) are trained on 0.66B training tokens. The best-performed scores are marked as **Bold**, and the second-best scores are Underlined.

| Method | ARC-C | ARC-E | HellaSwag | LogiQA | PIQA | SciQ || Average | Worst |
|--------|-------|-------|-----------|--------|------|------||---------|-------|
| Uniform | 4.83 | 4.92 | 4.30 | 4.77 | 4.08 | 4.56 || 4.58 | 4.92 |
| CRISP | 4.94 | 5.09 | 4.41 | 4.95 | 4.10 | 4.60 || 4.68 | 5.09 |
| REGMIX | 4.29 | 4.36 | 4.05 | **3.68** | 4.16 | **3.81** || 4.06 | 4.36 |
| DOGE | 4.35 | 4.51 | 3.83 | 4.34 | 3.66 | 4.09 || 4.13 | 4.51 |
| DOGE+PCGRAD | 4.38 | 4.54 | 3.88 | 4.35 | 3.71 | 4.18 || 4.17 | 4.54 |
| GRAPE | **4.05** | **4.08** | **3.23** | 4.19 | **3.54** | 4.04 || **3.86** | **4.19** |

Table 6: **5-shot accuracies(%) on target reasoning tasks**(↑). 125M (resp. 0.7B) models are trained on 0.66B (resp. 6.5B) tokens. The best-performed scores are marked as **Bold**, and the second-best scores are Underlined.

| 125M | ARC-C | ARC-E | HellaSwag | LogiQA | PIQA | SciQ | Average |
|---|---|---|---|---|---|---|---|
| Uniform | 21.50 | 32.45 | 26.44 | 26.42 | 55.50 | 52.40 | 35.78 |
| CRISP | 21.08 | 30.89 | 26.58 | 27.80 | 54.62 | 53.80 | 35.80 |
| REGMIX | 20.56 | 32.79 | **26.83** | 24.42 | **58.92** | 52.50 | 36.00 |
| DOGE | 21.67 | 32.66 | 26.05 | **30.11** | 57.18 | 56.60 | 37.38 |
| DOGE-PCGRAD | 21.33 | 32.20 | 25.96 | 27.19 | 58.60 | 53.50 | 36.46 |
| GRAPE | **21.76** | **34.09** | 26.69 | 26.27 | 58.60 | **58.50** | **37.65** |

| 0.7B | ARC-C | ARC-E | Hellaswag | PIQA | SciQ | LogiQA | Average |
|---|---|---|---|---|---|---|---|
| Uniform | 21.84 | 37.84 | 29.04 | 62.13 | 69.60 | **29.34** | 41.63 |
| DOGE | 22.87 | 40.61 | 31.16 | 63.82 | **69.80** | 24.88 | 42.19 |
| GRAPE | **23.46** | **42.17** | **31.90** | **64.42** | 68.70 | 27.50 | **42.92** |

**Reweighting with 8 Target Tasks: ARC-E, ARC-C, Hellaswag, SciQ, PIQA, LogiQA, MathQA, MedQA.** We present the full results on multi-task reasoning experiment, where the data mixture from SlimPajama dataset are adapted towards the optimal performance on 8 target tasks: ARC-E, ARC-C, Hellaswag, SciQ, PIQA, LogiQA, MathQA, MedQA, covering a board topics and knowledge domains. The Log-perplexity and 5-shot accuracy scores on each target reasoning tasks are presented in Figure 15, Table 8 and Table 7, where GRAPE consistently outperforms the other 5 baselines on most of benchmarks (6 out of 8), while achieving comparable performance as RegMix and DoGE-PCGrad on Hellaswag and PIQA. Notably, GRAPE improves the perplexity scores on MedQA by a large margin above all the other baselines, indicating its distinct adaptability to unique target tasks with domain-specific skill and knowledge. In contrast, CRISP demonstrates marginal improvements compared to uniform sampling baseline, which demonstrates its weakness in the multi-task learning setting.

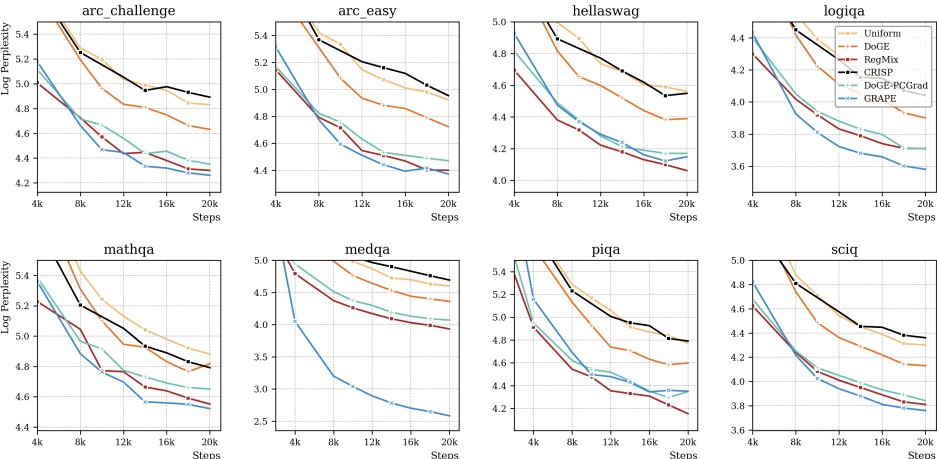

Figure 15: Log-Perplexities on 8 target datasets

Table 7: **5-shot accuracies(%) on target tasks.** The best-performed scores are marked as **Bold**, and the second-best scores are Underlined.

| Method | ARC-C | ARC-E | HellaSwag | LogiQA | PIQA | SciQ | MathQA | MedQA ‖ | Average |
|---|---|---|---|---|---|---|---|---|---|
| Uniform | 21.50 | 32.45 | 26.44 | 26.42 | 55.50 | 52.40 | 20.97 | 24.59 | 32.53 |
| CRISP | **22.01** | 30.56 | 26.24 | 27.19 | 55.88 | 51.10 | 19.97 | 21.21 | 29.27 |
| REGMIX | 22.00 | 32.40 | **26.80** | 25.50 | **59.70** | 56.00 | 20.70 | 21.40 | 33.10 |
| DOGE | 21.59 | 31.57 | 26.32 | 25.96 | 56.31 | 48.60 | 19.50 | **27.34** | 32.14 |
| DOGE+PCGRAD | 21.16 | 32.28 | 26.17 | 26.11 | 57.62 | 52.2 | 20.60 | 21.52 | 32.21 |
| GRAPE | 21.76 | **33.08** | 26.20 | **27.34** | 57.70 | **57.94** | **20.74** | 27.18 | **33.99** |

Table 8: **Log perplexity scores on 8 target datasets** ($\downarrow$). Models (125M) are trained on 0.66B training tokens. The best-performed scores are marked as **Bold**, and the second-best scores are Underlined.

| Method | ARC-C | ARC-E | HellaSwag | LogiQA | PIQA | SciQ | MathQA | MedQA ‖ | Average | Worst |
|---|---|---|---|---|---|---|---|---|---|---|
| Uniform | 4.83 | 4.92 | 4.56 | 4.08 | 4.77 | 4.30 | 4.88 | 4.60 | 4.62 | 4.92 |
| CRISP | 4.89 | 4.95 | 4.55 | 4.04 | 4.79 | 4.36 | 4.79 | 4.69 | 4.63 | 4.95 |
| REGMIX | 4.3 | 4.4 | **4.06** | 3.71 | **4.15** | 3.81 | 4.55 | 3.93 | 4.11 | 4.55 |
| DOGE+PCGRAD | 4.35 | 4.47 | 4.17 | 3.71 | 4.35 | 3.84 | 4.65 | 4.07 | 4.20 | 4.65 |
| DOGE | 4.63 | 4.72 | 4.39 | 3.90 | 4.60 | 4.13 | 4.82 | 4.36 | 4.44 | 4.82 |
| GRAPE | **4.26** | **4.37** | 4.15 | **3.58** | 4.35 | **3.76** | **4.52** | **2.58** | **3.95** | **4.52** |

## D.4 Results on Wiki-40b

We present detailed results on the multilingual pretraining experiments on Wiki-40B dataset in the following sections. Specifically, we experiment with two various target configurations: (1) target at 4 low-resource languages: Catalan, Danish, Romanian, Ukrainian; (2) target at 8 low-resource languages: Catalan, Danish, Romanian, Ukrainian, Polish, Portuguese, Turkish, Dutch, i.e. the setting we present in the main paper (§ 3.2).

### D.4.1 Language Mixture from Various Reweighting Algorithms

**Domain Weights from RegMix.** We present the optimized domain weights distributions from RegMix in Figure 16 and 17. Specifically, with 4 target languages, RegMix upweights German (de) and Spanish (es) while reducing the weights assigned on French (fr) and Italian (it). With 8 target languages, RegMix significantly increases the domain weights on Russian (ru), while further decreases the proportion of English (en).

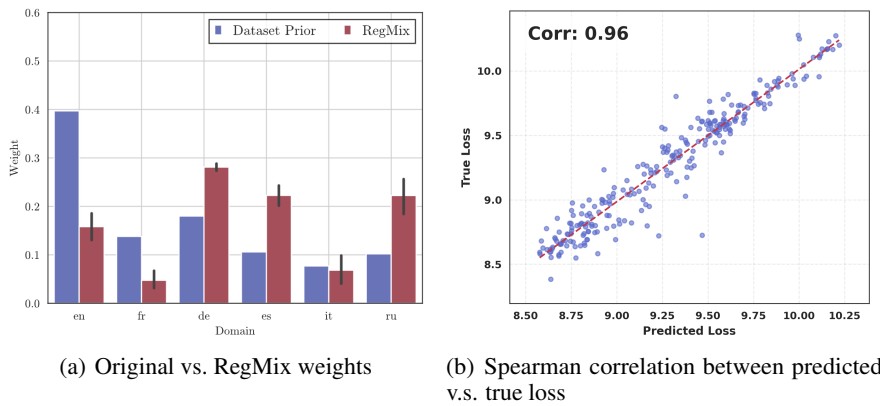

(a) Original vs. RegMix weights

(b) Spearman correlation between predicted v.s. true loss

Figure 16: **RegMix results on `wiki-40b` with LightGBM regression on 4 target languages** [Catalan, Danish, Romanian, Ukrainian]

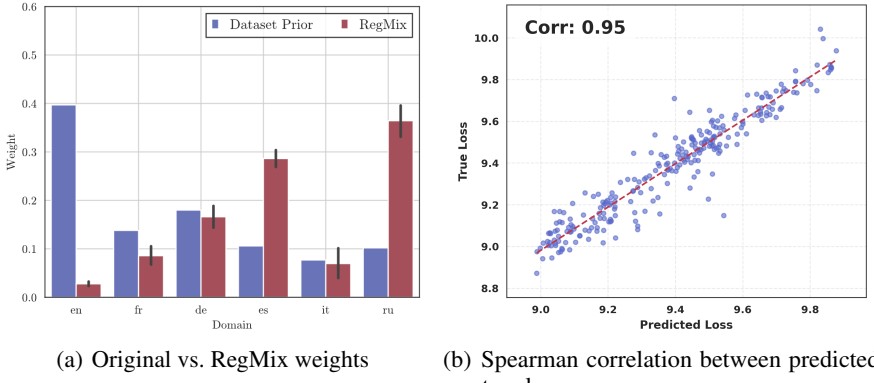

(a) Original vs. RegMix weights

(b) Spearman correlation between predicted v.s. true loss

Figure 17: **RegMix results on `wiki-40b` with LightGBM regression on 8 target languages** [Catalan, Danish, Romanian, Ukrainian, Polish, Portuguese, Turkish, Dutch]

**Domain Weights from CRISP.** We present the optimized domain weights distributions from CRISP in Figure 18, with various task configurations. Notably, CRISP yields very similar domain weights across 6 languages, where English (en) and Italian (it) are mostly upweighted, while they are proved not helpful for multi-language learning according to Figure 19 and 20.

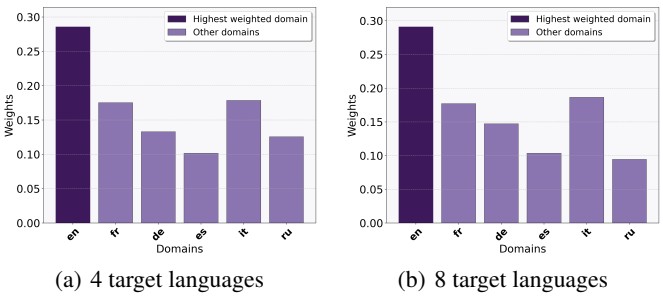

(a) 4 target languages

(b) 8 target languages

Figure 18: CRISP Domain Weigths across 6 high-resource languages in `wiki-40b`

### D.4.2 Results on Various Task Configurations

**Reweighting with 4 target languages: Catalan, Danish, Romanian, Ukrainian** Figure 19 present the full results on the multilingual pretraining experiments, where the training langauge mixture are adapted to 4 target low-resource languages: Catalan, Danish, Romanian, Ukrainian. GRAPE significantly outperforms all the other baseline methods across all target languages. Notably, none of the other reweighting approaches can effectively facilitate the learning on Romanian, while only GRAPE unleashes the learning capability on it.

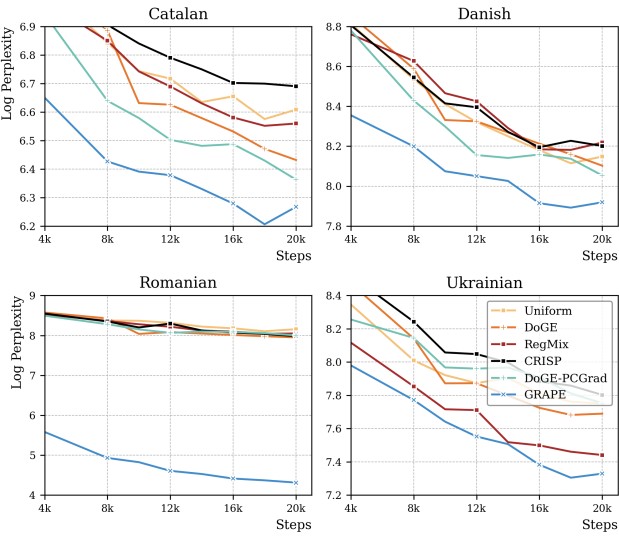

Figure 19: **Log-perplexities on 4 target languages** [Catalan, Danish, Romanian, Ukrainian]

**Reweighting with 8 target languages: Catalan, Danish, Romanian, Ukrainian, Polish, Portuguese, Turkish, Dutch.** Figure 20 present the full results on the multilingual pretraining experiments, where the training langauge mixture are adapted to 8 target low-resource languages: Catalan, Danish, Romanian, Ukrainian, Polish, Portuguese, Turkish, Dutch. GRAPE significantly outperforms all the other baseline methods across all 8 target languages. In contrast, the offline reweighting algorithm such as RegMix and CRISP exhibit a very biased performance across various target languages: RegMix accelerate the learning on Catalan, Portuguese, Ukrainian and Polish, while sacrificing the performance on Dutch and Danish; CRISP only accelerates the learning on Turkish while sabotaging all the other languages.

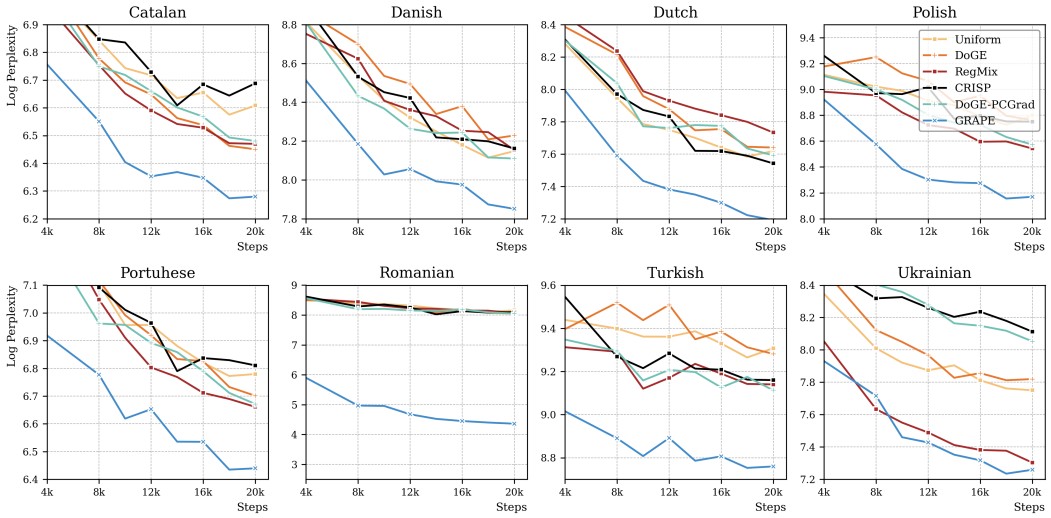

Figure 20: **Log-perplexities on 8 target languages** [Catalan, Danish, Romanian, Ukrainian, Polish, Portuguese, Turkish, Dutch]

# E   Ablation Study

In this section, we present the full results on our ablation experiments, with three various progress measurement metrics for Group DRO, and 12 different target task configurations.

## E.1   Progress Assessment Metrics for Group DRO

We evaluate the impact of different step-wise progress metrics on the performance of Group DRO within the GRAPE framework. In addition to the primary Rate-of-Improvement (ROI) metric (Equation 1), we investigate two alternatives: *Gap-of-Improvement* (*GOI*) and *EMA-Rate-of-Improvement* (*ROI-ema*). The step-wise improvmenet of task $\mathcal{T}_n$ at step $t$ are assessed as:

$$GOI_n^{(t)} := l_n(\boldsymbol{\theta}_t) - l_n(\boldsymbol{\theta}_{t+1}), \quad ROI\text{-}ema_n^{(t)} := \frac{l_n(\boldsymbol{\theta}_t) - l_n(\boldsymbol{\theta}_{t+1})}{l_{ema,n}^t}, \qquad \forall n \in [N] \quad (7)$$

The exponential moving average loss $l_{ema,n}^t$ is updated as: $l_{ema,n}^t = \beta \cdot l_{ema,n}^{t-1} + (1-\beta) \cdot l_n(\boldsymbol{\theta}_t)$, where the hyperparameter $\beta$ is set to $0.7$ in our experiments. Substituting these alternative progress metrics for $r_n$ in the minimax objective Equation 2 yields modified update rules according to Equation 8 and 9. The core optimization principle for adjusting task weights ($\boldsymbol{z}$) and domain weights ($\boldsymbol{\alpha}$) remains, but the specific gradient terms within the exponents change.

- *GOI*:

$$\begin{cases} \boldsymbol{z}^t = \frac{\hat{\boldsymbol{z}}^t}{\sum_{n\in[N]} \hat{z}_n^t}, & \text{with } \hat{z}_n^t \leftarrow z_n^{t-1} \cdot \exp\left( -\frac{\gamma_t}{\mu_{\boldsymbol{z}}} \mathbb{E}_{\boldsymbol{x}\sim\text{mix}(\boldsymbol{\alpha}^{t-1})}[\langle \nabla_{\boldsymbol{\theta}} l_n(\boldsymbol{\theta}_t), \nabla_{\boldsymbol{\theta}} \ell(\boldsymbol{\theta}_t, \boldsymbol{x}) \rangle] \right), \\ \boldsymbol{\alpha}^t = \frac{\hat{\boldsymbol{\alpha}}^t}{\sum_{k\in[K]} \hat{\alpha}_k^t}, & \text{with } \hat{\alpha}_k^t \leftarrow \alpha_k^{t-1} \cdot \exp\left( \frac{\gamma_t}{\mu_{\boldsymbol{\alpha}}} \mathbb{E}_{\boldsymbol{y}\sim\text{mix}(\boldsymbol{z}^{t-1})}[\langle \boldsymbol{g}_k(\boldsymbol{\theta}_t), \nabla_{\boldsymbol{\theta}} \ell(\boldsymbol{\theta}_t, \boldsymbol{y}) \rangle] \right). \end{cases} \quad (8)$$

- *ROI-ema*:

$$\begin{cases} \boldsymbol{z}^t = \frac{\hat{\boldsymbol{z}}^t}{\sum_{n\in[N]} \hat{z}_t^t}, & \text{with } \hat{z}_n^t \leftarrow z_n^{t-1} \cdot \exp\left( -\frac{\gamma_t}{\mu_{\boldsymbol{z}}} \mathbb{E}_{\boldsymbol{x}\sim\text{mix}(\boldsymbol{\alpha}^{t-1})}[\langle \frac{\nabla_{\boldsymbol{\theta}} l_n(\boldsymbol{\theta}_t)}{l_{ema,n}^t}, \nabla_{\boldsymbol{\theta}} \ell(\boldsymbol{\theta}_t, \boldsymbol{x}) \rangle] \right), \\ \boldsymbol{\alpha}^t = \frac{\hat{\boldsymbol{\alpha}}^t}{\sum_{k\in[K]} \hat{\alpha}_k^t}, & \text{with } \hat{\alpha}_k^t \leftarrow \alpha_k^{t-1} \cdot \exp\left( \frac{\gamma_t}{\mu_{\boldsymbol{\alpha}}} \mathbb{E}_{\boldsymbol{y}\sim\text{mix}(\boldsymbol{z}^{t-1})}[\langle \boldsymbol{g}_k(\boldsymbol{\theta}_t), \nabla_{\boldsymbol{\theta}} \log \ell(\boldsymbol{\theta}_t, \boldsymbol{y}) \rangle] \right). \end{cases} \quad (9)$$

**Task Configurations.**   We evaluate the efficacy of GRAPE on 12 various target task combinations as follows:

- T1: GSM8K, ARC-C, ARC-E
- T2: GSM8K, Hellaswag
- T3: GSM8K, PIQA
- T4: GSM8K, LogiQA
- T5: GSM8K, SciQ
- T6: GSM8K, ARC-E, ARC-C, Kodcode
- T7: GSM8K, Kodcode, Hellaswag
- T8: ARC-E, ARC-C, Hellaswag, SciQ, PIQA, LogiQA, Kodcode, GSM8K
- T9: ARC-E, ARC-C, Hellaswag, SciQ, PIQA, LogiQA
- T10: ARC-E, ARC-C, Hellaswag, SciQ, PIQA, LogiQA, MathQA, MedQA
- T11: ARC-E, ARC-C, MathQA, MedQA
- T12: LogiQA, Hellaswag, MathQA, MedQA

**Results.** We present the average and worst log-perplexity scores on each of 12 groups of ablation experiments in Table 9. Specifically, GRAPE outperforms uniform and DoGE across all task sets; on 7 out of 12 tasks, GRAPE with $ROI$ metric outperforms the other two DRO variants with $GOI$ and $ROI\text{-}ema$, respectively. We present the log-perplexity scores associated with task weights evolution trajectories for all 12 groups of ablations in Figure 21-32.

Table 9: **Average and worst-case test log-perplexity($\downarrow$) on 12 task combinations (trained on 0.66 B** `SlimPajama`**) tokens.** GRAPE outperforms uniform and DoGE across all task sets; on 7 out of 12 tasks, GRAPE with $ROI$ metric outperforms the other two DRO variants.

| Method (avg. \| wst.) | Uniform | DOGE | GRAPE | GRAPE-ema | GRAPE-gap |
|---|---|---|---|---|---|
| T1 | 4.30 \| 4.66 | 4.18 \| 4.53 | **3.53 \| 3.55** | 3.68 \| 3.73 | 3.58 \| 3.59 |
| T2 | 4.01 \| 4.35 | 3.94 \| 4.24 | **3.61 \| 3.70** | 3.70 \| 3.81 | 3.68 \| 3.70 |
| T3 | 4.11 \| 4.55 | 4.00 \| 4.38 | 3.70 \| 3.85 | 3.73 \| 3.89 | **3.69 \| 3.80** |
| T4 | 3.75 \| 3.83 | 3.67 \| 3.69 | **3.40 \| 3.47** | 3.38 \| 3.46 | 3.41 \| 3.54 |
| T5 | 3.85 \| 4.02 | 3.70 \| 3.85 | 3.37 \| **3.43** | 3.37 \| 3.47 | **3.35** \| 3.48 |
| T6 | 3.62 \| 4.66 | 3.63 \| 4.77 | **3.04** \| 3.58 | 3.10 \| 3.65 | 3.05 \| **3.52** |
| T7 | 3.20 \| 4.34 | 3.12 \| 4.26 | **2.93** \| 3.80 | 2.96 \| 3.81 | 3.11 \| **3.72** |
| T8 | 3.90 \| 4.66 | 3.58 \| 4.58 | 3.46 \| **4.30** | **3.41** \| 4.42 | 3.51 \| 4.32 |
| T9 | 4.33 \| 4.66 | 4.13 \| 4.51 | **3.86 \| 4.19** | 3.88 \| 4.21 | 3.88 \| 4.21 |
| T10 | 4.62 \| 4.92 | 4.32 \| 4.82 | 3.95 \| 4.52 | **3.66 \| 4.51** | 3.86 \| 4.55 |
| T11 | 4.81 \| 4.92 | 4.57 \| 4.74 | 3.94 \| **4.52** | 3.90 \| 4.53 | **3.85** \| 4.53 |
| T12 | 4.53 \| 4.88 | 4.33 \| 4.76 | 3.70 \| 4.51 | 3.73 \| 4.56 | **3.59 \| 4.50** |

## E.2 Task Combination 1: GSM8K, ARC-C, ARC-E

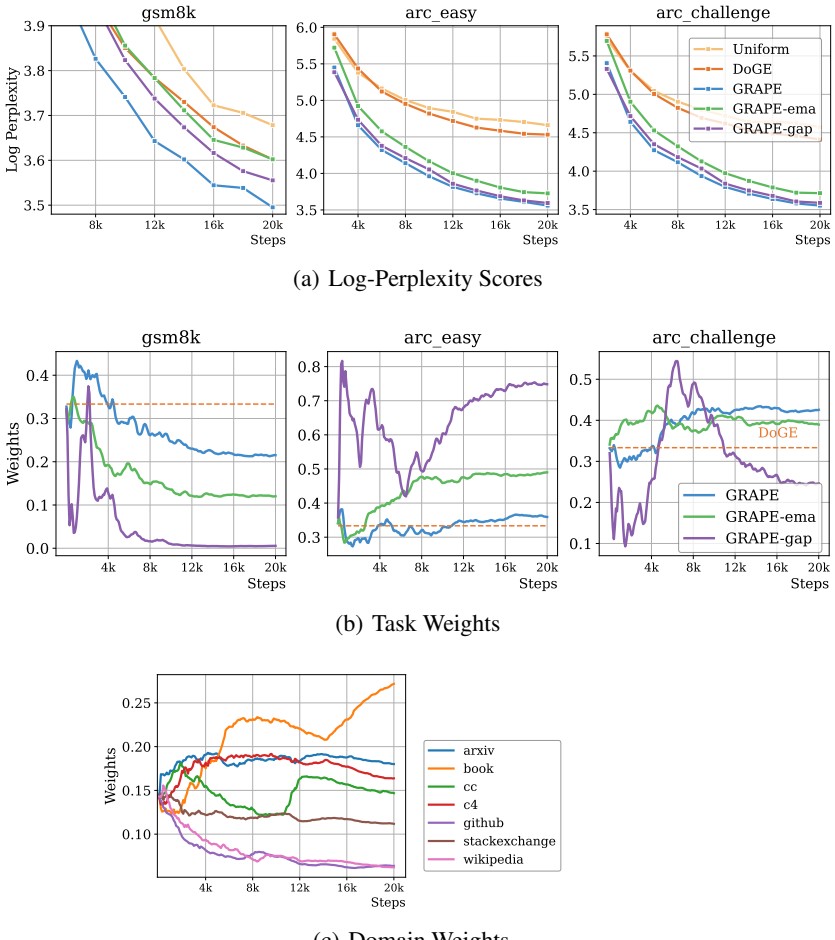

(a) Log-Perplexity Scores

(b) Task Weights

(c) Domain Weights

Figure 21: **Ablation on task combination [GSM8K, ARC-C, ARC-E].**

## E.3 Task Combination 2: GSM8K, Hellaswag

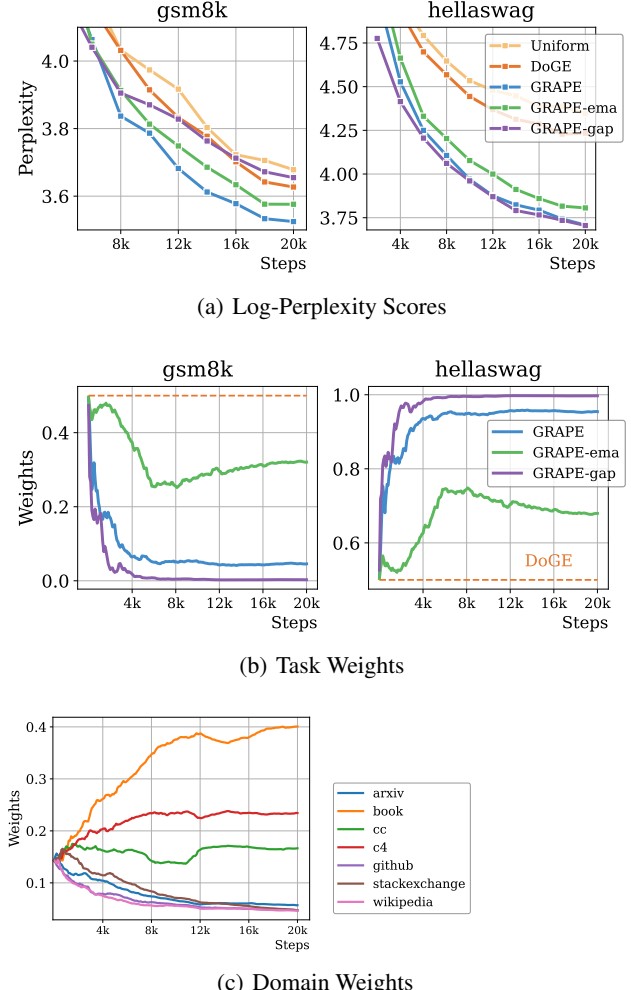

(a) Log-Perplexity Scores

(b) Task Weights

(c) Domain Weights

Figure 22: **Ablation on task combination [GSM8K, Hellaswag].**

## E.4   Task Combination 3: GSM8K, PIQA

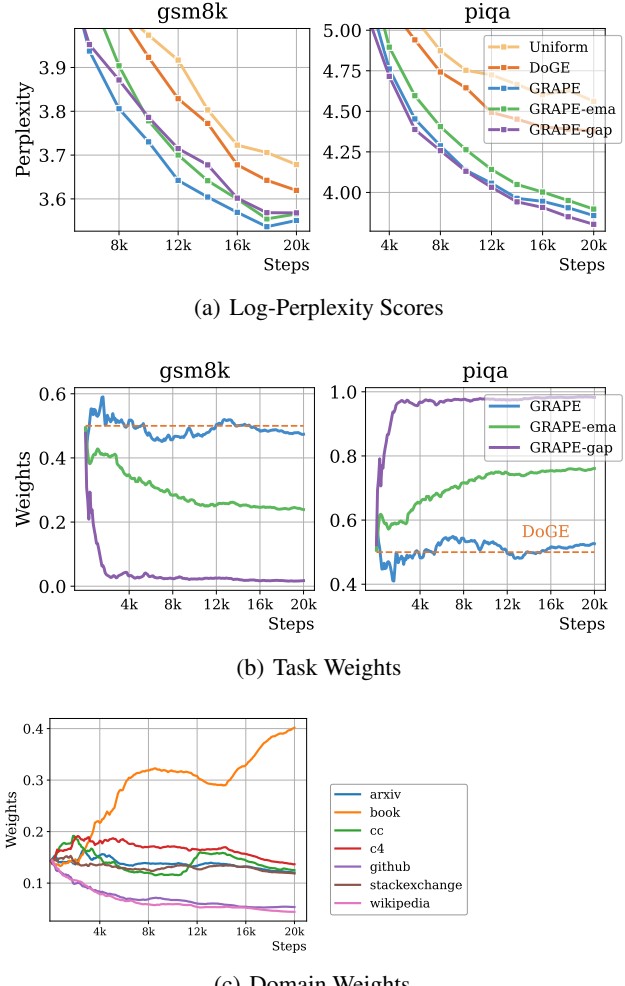

(a) Log-Perplexity Scores

(b) Task Weights

(c) Domain Weights

Figure 23: **Ablation on task combination [GSM8K, PIQA].**

## E.5 Task Combination 4: GSM8K, LogiQA

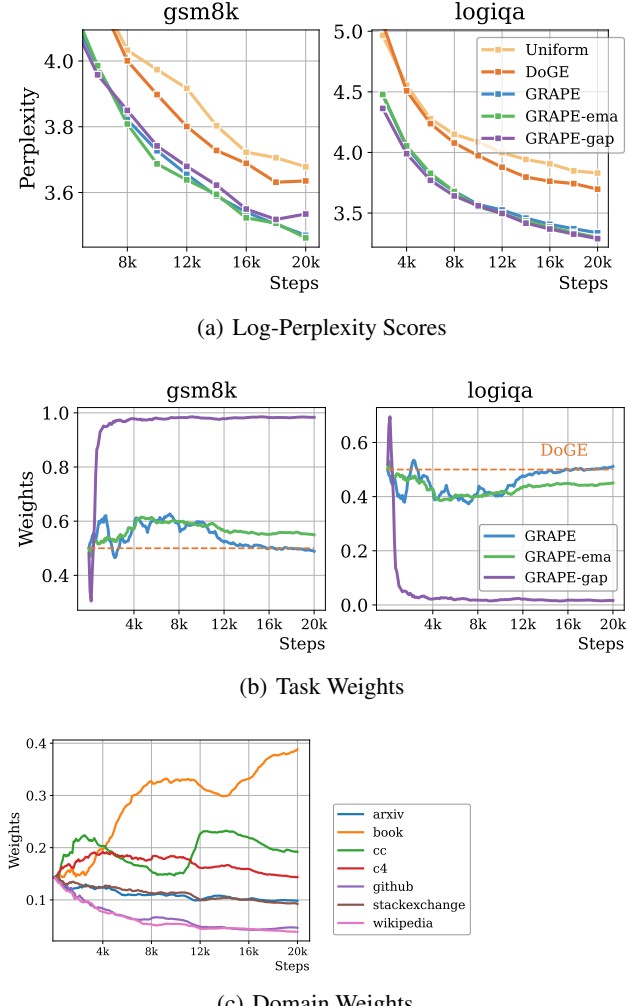

(a) Log-Perplexity Scores

(b) Task Weights

(c) Domain Weights

Figure 24: **Ablation on task combination [GSM8K, LogiQA].**

## E.6 Task Combination 5: GSM8K, SciQ

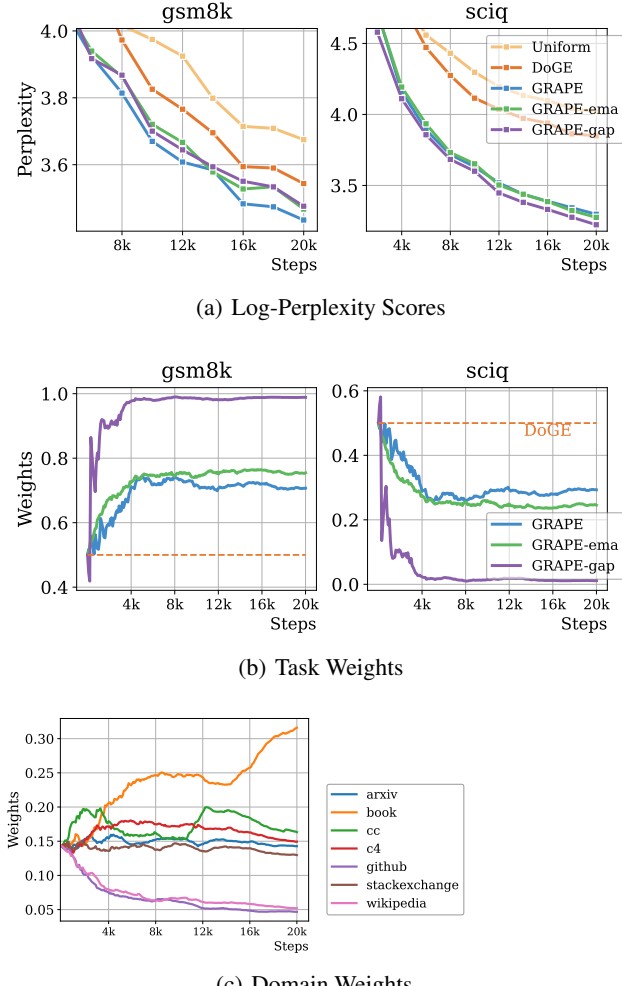

(a) Log-Perplexity Scores

(b) Task Weights

(c) Domain Weights

Figure 25: **Ablation on task combination [GSM8K, SciQ].**

## E.7 Task Combination 6: GSM8K, ARC-E, ARC-C, Kodcode

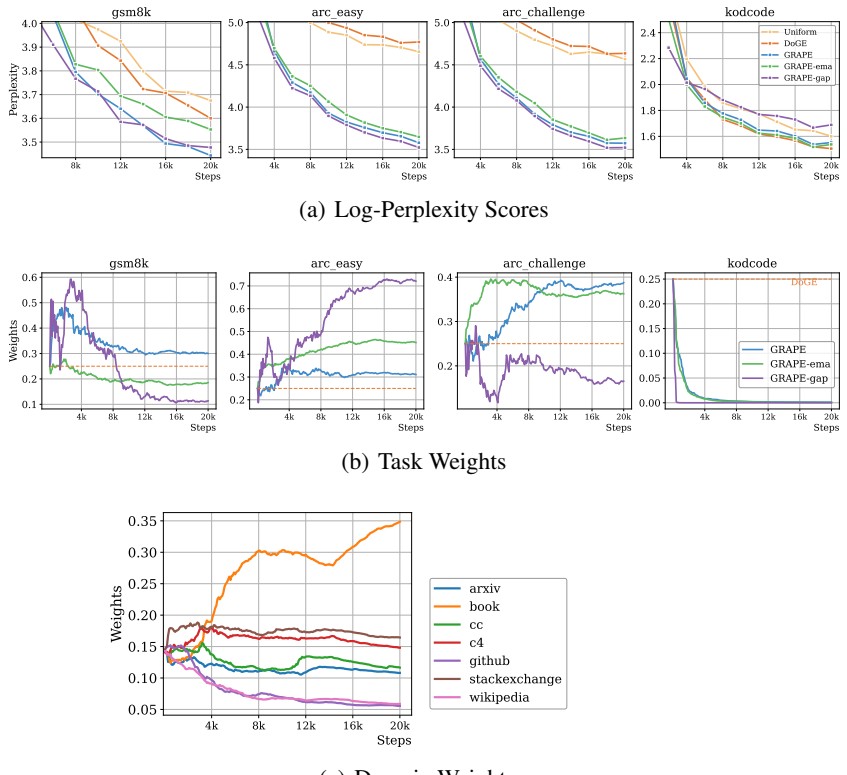

(a) Log-Perplexity Scores

(b) Task Weights

(c) Domain Weights

Figure 26: **Ablation on task combination [GSM8K, ARC-E, ARC-C, Kodcode].**

## E.8 Task Combination 7: GSM8K, Kodcode, Hellaswag

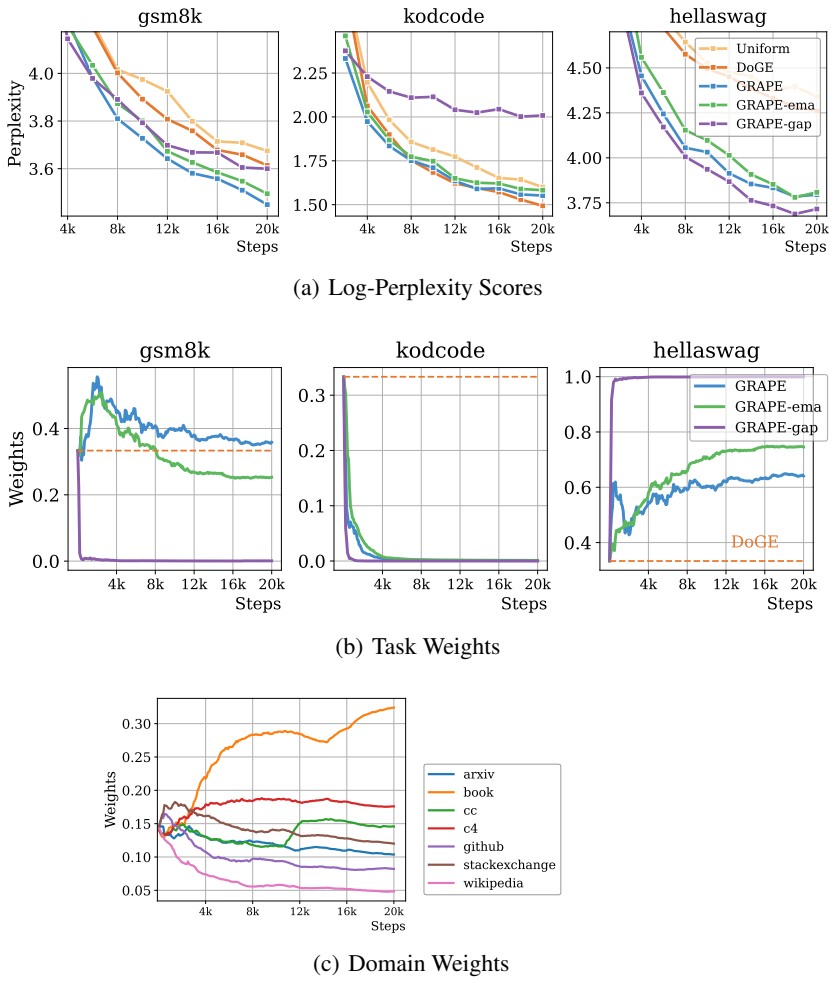

(a) Log-Perplexity Scores

(b) Task Weights

(c) Domain Weights

Figure 27: **Ablation on task combination [GSM8K, Kodcode, Hellaswag].**

## E.9 Task Combination 8: ARC-E, ARC-C, Hellaswag, SciQ, PIQA, LogiQA, Kodcode, GSM8K

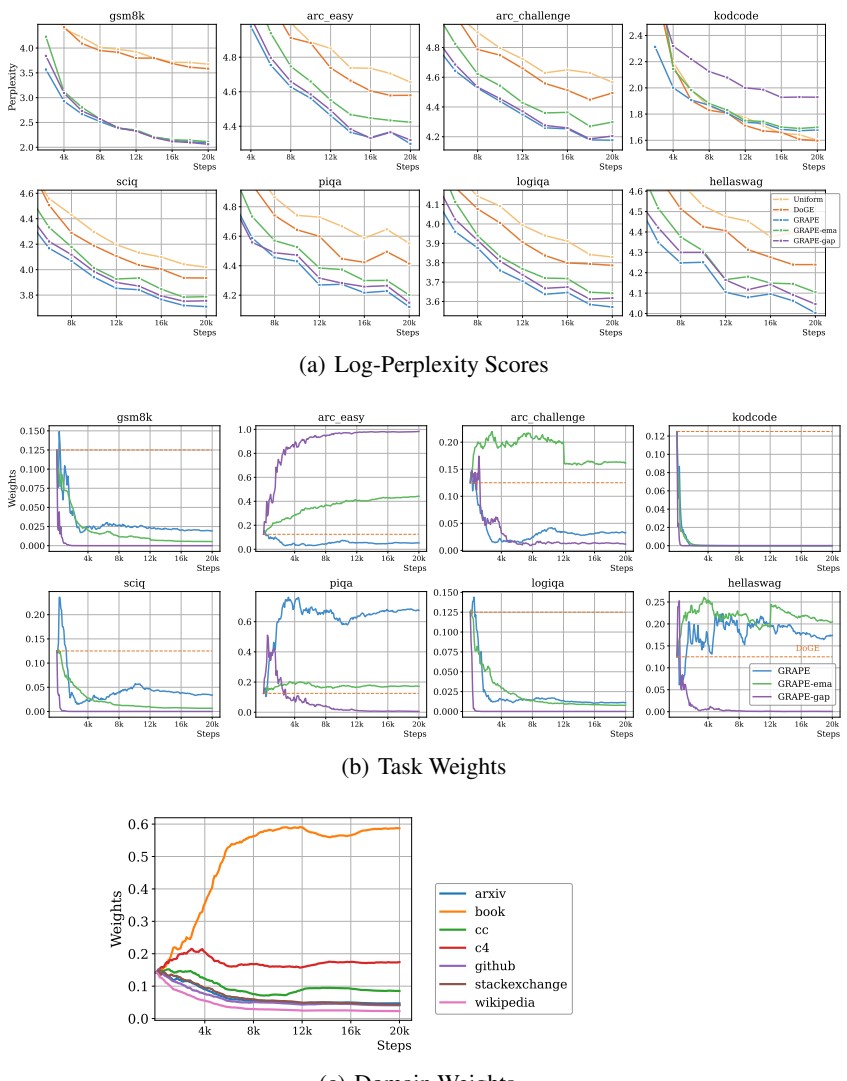

(a) Log-Perplexity Scores

(b) Task Weights

(c) Domain Weights

Figure 28: **Ablation on task combination [ARC-E, ARC-C, Hellaswag, SciQ, PIQA, LogiQA, Kodcode, GSM8K].**

## E.10 Task Combination 9: ARC-E, ARC-C, Hellaswag, SciQ, PIQA, LogiQA

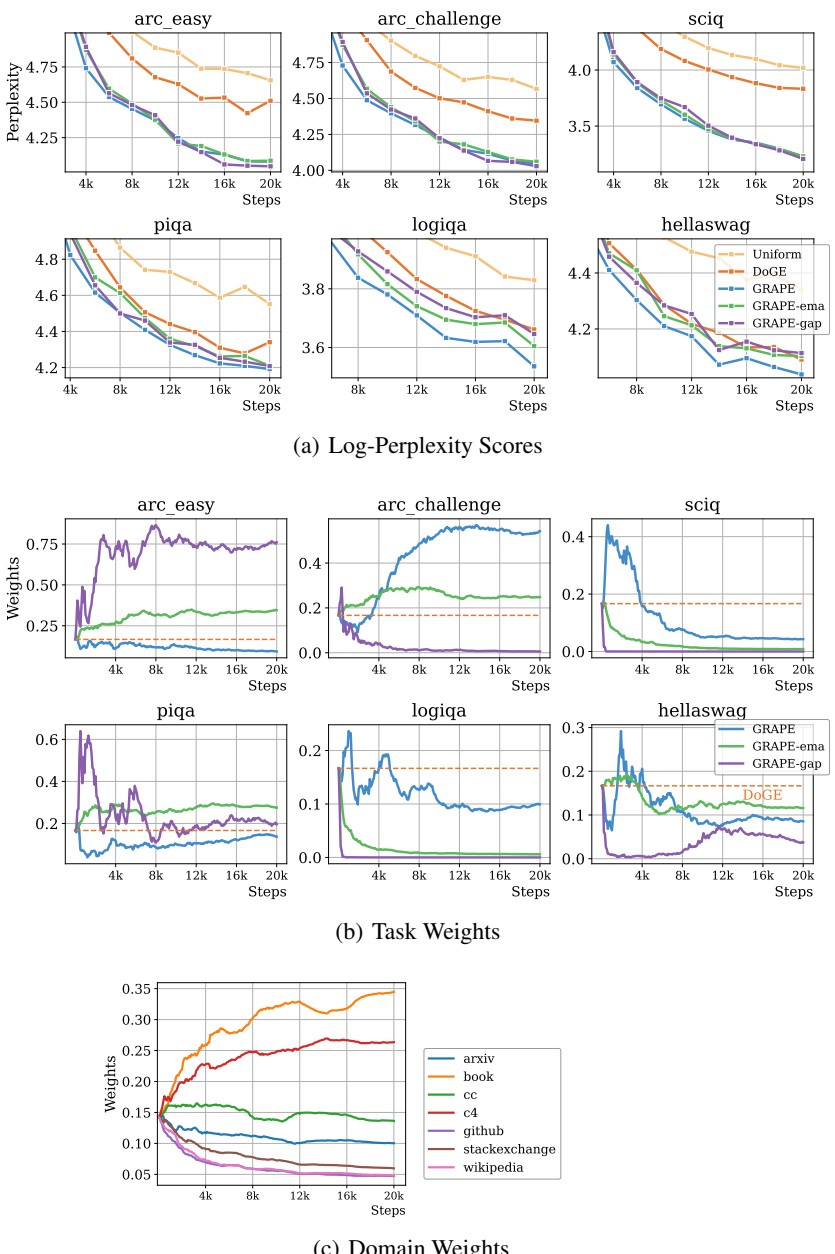

(a) Log-Perplexity Scores

(b) Task Weights

(c) Domain Weights

Figure 29: **Ablation on task combination [ARC-E, ARC-C, Hellaswag, SciQ, PIQA, LogiQA].**

### E.11 Task Combination 10: ARC-E, ARC-C, Hellaswag, SciQ, PIQA, LogiQA, MathQA, MedQA

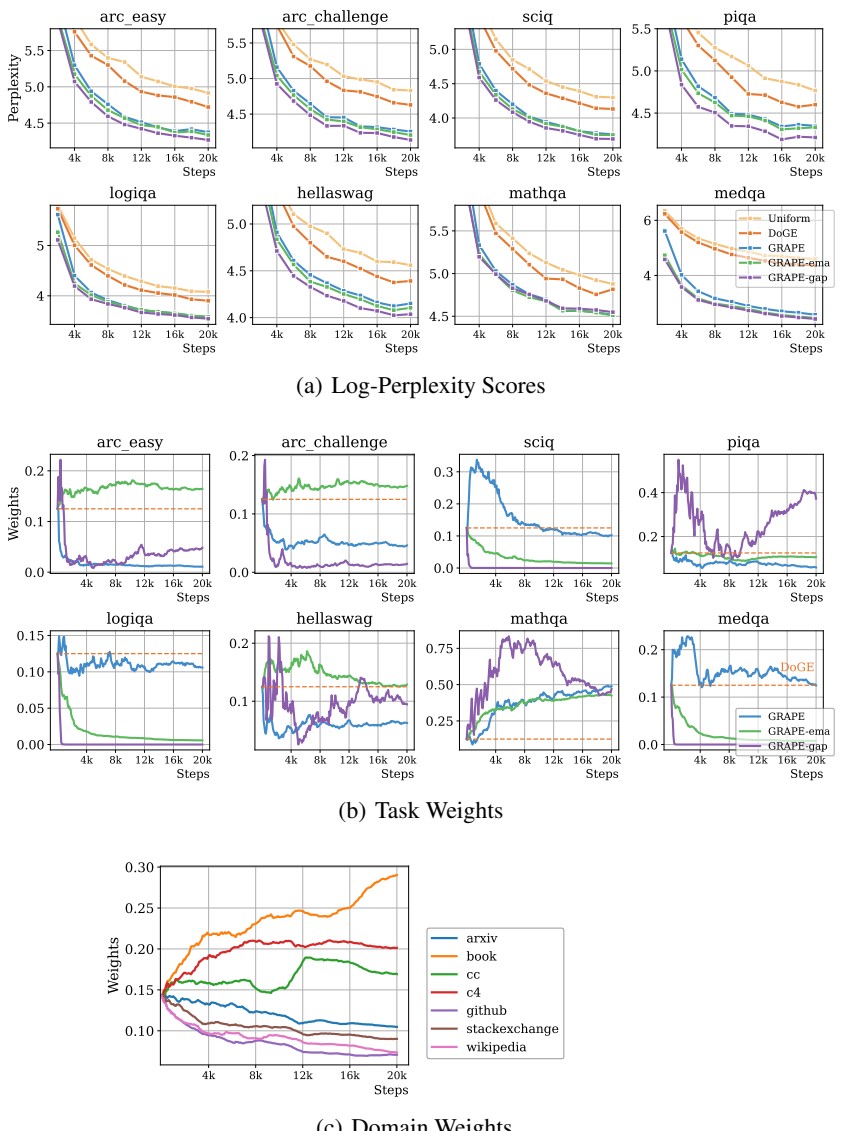

(a) Log-Perplexity Scores

(b) Task Weights

(c) Domain Weights

Figure 30: **Ablation on task combination [ARC-E, ARC-C, Hellaswag, SciQ, PIQA, LogiQA, MathQA, MedQA].**

## E.12 Task Combination 11: ARC-E, ARC-C, MathQA, MedQA

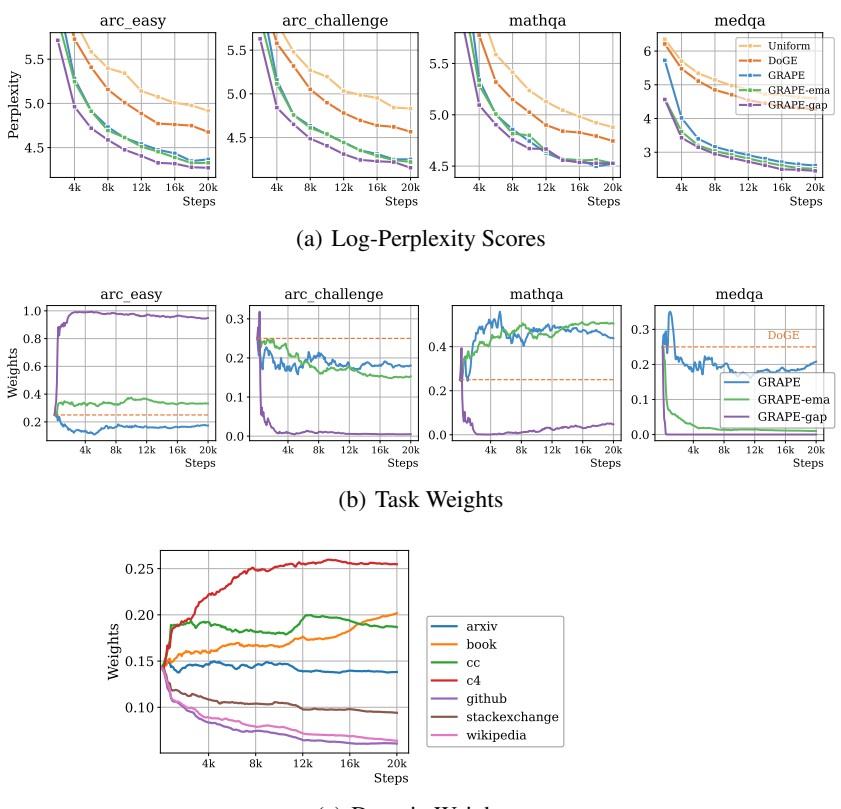

(a) Log-Perplexity Scores

(b) Task Weights

(c) Domain Weights

Figure 31: **Ablation on task combination [ARC-E, ARC-C, MathQA, MedQA].**

### E.13 Task Combination 12: LogiQA, Hellaswag, MathQA, MedQA

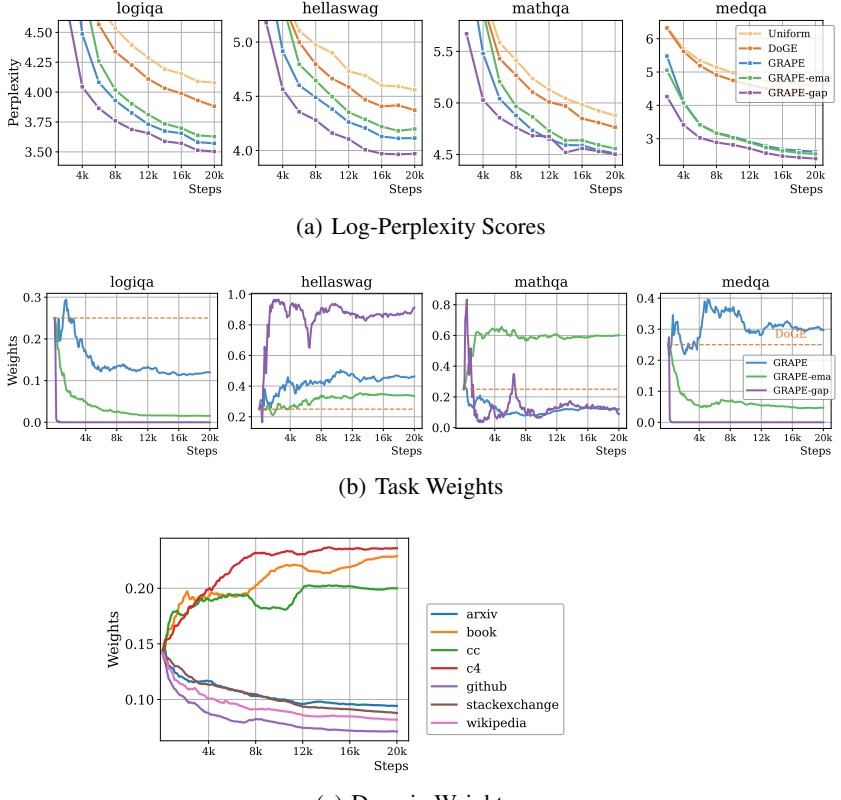

(a) Log-Perplexity Scores

(b) Task Weights

(c) Domain Weights

Figure 32: **Ablation on task combination [LogiQA, Hellaswag, MathQA, MedQA].**

## F  Hyperparameter Tuning

We perform a hyperparameter search on the reweighting intervals $\Delta T_\alpha$, $\Delta T_\alpha = 50, 100, 200$ for both GRAPE and DoGE. We train $125M$ model for $50k$ steps and report the final task accuracies and run time as in Table 10. These results show that GRAPE consistently outperforms DoGE across all tested update frequencies. We also observe that the most frequent updates ($\Delta T_\alpha$=50) do not yield the best performance. This is likely because the Rate-of-Improvement (RoI) metric becomes more reliable after the model has trained for a sufficient number of steps on the current data mixture, making overly frequent weight adjustments suboptimal.

Table 10: **Hyperparameter Tuning on Reweighting Frequency and Efficiency Comparison on GRAPE and DoGE.** GRAPE with $\Delta T_\alpha$=$\Delta T_z$=100 outperforms 50 and 200.

| $\Delta T_\alpha, \Delta T_\alpha$ | Method | ARC-Challenge | ARC-Easy | Hellaswag | Logiqa | PIQA | SciQ | Average | Runtime (H100) |
|---|---|---|---|---|---|---|---|---|---|
| 50 | GRAPE | 25.09% | 45.83% | 27.86% | 25.50% | 60.12% | 73.30% | 42.95% | 23.9$h$ |
| | DoGE | 24.40% | 43.64% | 27.53% | 26.88% | 58.98% | 70.10% | 41.92% | 22.3$h$ |
| 100 | GRAPE | 26.11% | 47.14% | 28.56% | 27.65% | 61.10% | 74.40% | 44.16% | 15.8$h$ |
| | DoGE | 24.57% | 45.33% | 27.77% | 25.35% | 59.52% | 72.20% | 42.45% | 15.0$h$ |
| 200 | GRAPE | 23.81% | 44.40% | 27.77% | 29.03% | 59.58% | 72.70% | 42.88% | 10.3$h$ |
| | DoGE | 24.32% | 43.31% | 27.07% | 28.42% | 59.14% | 70.10% | 42.06% | 10.8$h$ |

