# OpenReview forum: "GRAPE: Optimize Data Mixture for Group Robust Multi-target Adaptive Pretraining"
_NeurIPS.cc/2025/Conference — NeurIPS 2025 poster_

### Official Review · Reviewer_qr9c · 2025-06-08

**Clarity:** 4
**Significance:** 3
**Originality:** 3
**Rating:** 4
**Confidence:** 3

**Summary:**

This paper addresses the challenge of balancing training across multiple source and target domains in language model pretraining. It optimizes the Distributed Robust Optimization (DRO) algorithm by using the rate of improvement on test sets as a metric, rather than just training set loss. This allows the method to dynamically adjust task weights to prioritize more challenging tasks, and subsequently adjust domain weights to better achieve this learning objectives. The efficacy of this approach is demonstrated through experiments in both monolingual and multilingual environments.

**Questions:**

1. Generalization to Unseen Tasks: The primary objective of pretraining LLMs extends beyond achieving superior performance on a predefined set of tasks (six in the main experiments of this paper). LLMs are also expected to exhibit strong generalization capabilities. How does this training method perform on tasks not included in the training process, particularly those requiring different types of abilities or knowledge?
2. Applicability to Broader Scenarios: Can this method be effectively extended to broader practical applications, such as instruction tuning or continued pretraining?

**Ethical Concerns:**

["NO or VERY MINOR ethics concerns only"]

**Final Justification:**

The authors' paper is a complete piece of work and makes a meaningful contribution. Their response has mostly addressed my concerns. Regarding my second question, it is understandable that the authors could not complete the additional experiments within the limited rebuttal period. Therefore, considering the relatively positive score I have already given, I will maintain my rating.

**Limitations:**

Yes

**Quality:**

3

**Strengths And Weaknesses:**

Strengths:
1. Simplicity and Theoretical Soundness: The proposed method is straightforward and direct, backed by theoretical guarantees that support its overall effectiveness.
2. Clarity of Writing: The paper is well-written, making the methodology and results easy to understand.

Weaknesses:
1. Code Availability: The authors do not provide code for their implementation.
2. Limited Test Set Scope: The experiments utilize a relatively small number of test sets.

---

> ### Author Rebuttal · Authors · 2025-07-31
>
> We thank the reviewer for the thoughtful questions and constructive suggestions! We provide our response as follows:
> ## Q1: Generalizability to unseen tasks
> We evaluate the model optimized using the 6 reasoning tasks from our main experiment on 6 unseen tasks and report the 5-shot reasoning accuracy as follows:
> Method      	| RACE  | Winograd | OpenbookQA | MMLU  | CommonsenseQA | COPA
> ----------------|-------|----------|--------|-------|--------|-------
> Uniform     	| 25.65 | 49.80	| 15.40  | 25.07 | **20.15**  | 57.00
> DoGE        	| 26.12 | 49.80	| 26.40  | **26.43** | 19.82  | 60.00
> **Ours**    	| **27.37** | **52.49** | **28.60** | 25.53 | 18.92 | **61.00**
>
> Our method outperforms both the Uniform and the strong DoGE baseline on a majority (4 out of 6) of these unseen benchmarks, which suggests that GRAPE's advantages generalize well beyond the seen tasks.
>
>
>
>
> ## Q2: Broader applications to Post-training
> GRAPE is highly applicable as a general data mixture optimization method to broader scenarios, including continued pretraining and instruction tuning.
> During the post-training stage like instruction tuning, a more fine-grained data selection method can be preferable like a sample-level selection to get a better control of the final model behavior. Using sample-level DRO to upweight the most difficult individual instructions is a powerful concept for the SFT stage and a promising direction for future research, as we discussed in the Discussion section (line 353-359).
>
> We will add the above discussions to the revised version of the paper. For code availability, according to the rebuttal guideline, we cannot put external links here. We promise to open-source our codebase in the updated version.

---

> > ### Comment · Reviewer_qr9c · 2025-08-05
> > **Official Comment by Reviewer qr9c**
> >
> > Thanks for your response, which has mostly addressed my concerns.

---

### Official Review · Reviewer_qSR5 · 2025-06-21

**Clarity:** 4
**Significance:** 4
**Originality:** 3
**Rating:** 5
**Confidence:** 5

**Summary:**

This paper points out that pretraining has two goals: to reduce the loss on domains in the pretraining corpus, and to improve on downstream tasks. The authors introduce GRAPE, an algorithm that balances these two goals. GRAPE works similar to DOGE, using the DOGE update rule for both pretraining domain weights and downstream task weights.

**Questions:**

1. Do you have feedback for the reader on tuning hyperparameters? Hyperparameter tuning seems critical here because there are many of them.
2. What do you think of other data mixing methods like Data Mixing Laws, Aioli, and Adaptive Data Optimization? These methods aren't derived from the DRO framework. I'd be interested in seeing a little commentary what the benefit of the minimax DRO framework is, over just a minimization framework seen in the methods above and RegMix.
3. Some information about wall-clock time would be helpful. Although GRAPE is an online method, it will take longer than normal pretraining because you're taking lots of gradients on validation sets.
4. Relatedly, how would GRAPE interact with pretraining corpora that have many partitions, like DCLM-Baseline or WebOrganizer? In these cases it seems like you would have many validation gradients, in which case you would be decreasing your throughput because you have to allocate more of each batch to computing validation gradients.

**Ethical Concerns:**

["NO or VERY MINOR ethics concerns only"]

**Final Justification:**

Technically solid work. The authors adequately addressed my questions and provided creative and thought-provoking solutions. I have published recently in this area and am familiar with the existing literature, and this paper is an original departure from previous work. Thus, I maintain my score of accept and confidence of 5.

**Limitations:**

Yes

**Quality:**

4

**Strengths And Weaknesses:**

Strengths: The core points of the paper---that pretraining aims to improve performance on many downstream tasks, and that existing algorithms don't explicitly do this---are well made. The authors also make thoughtful evaluations on the curriculum and multilingual training. The results are strong and GRAPE seems to work. I think GRAPE is a well-motivated and natural evolution of DoReMi and DoGE.

Weaknesses: In using DoGE update rule twice, the GRAPE algorithm naturally more hypeparameters than DoGE. This can potentially be a challenge for hyperparameter turning.

Quality: The paper is impressively thorough.
Clarity: I found the paper easy to follow. I clearly understood both GRAPE's connection to and clear separation from prior work.
Significance: To my knowledge, this is the first algorithm that adjusts both domain and downstream task weights.
Originality: Ideas build on prior work but are applied to a new setting.

---

> ### Author Rebuttal · Authors · 2025-07-31
>
> We thank the reviewer for the thoughtful questions and constructive suggestions! We provide our response as follows:
>
> ## Q1: Hyperparameter tuning & Runtime:
> We perform a hyperparameter search on the reweighting intervals $\Delta T_{\alpha}, \Delta T_{z}$ = 50, 100, 200 for both GRAPE and DoGE. We train 125M model for 50k steps and report the final task accuracies and run time as follows:
>
> $\Delta Τ_{\alpha}, \Delta Τ_{z}$ | Method | arc_challenge | arc_easy | hellaswag | logiqa | piqa | sciq | Average | Runtime (H100)
> ----|--------|---------------|----------|-----------|--------|-------|-------|---------|--------
> 50 | GRAPE | 25.09% | 45.83% | 27.86% | 25.50%| 60.12%| 73.30%| 42.95% | 23.9h
> 50| DoGE | 24.40% | 43.64% | 27.53% | 26.88%| 58.98%| 70.10%| 41.92% | 22.3h
> 100 | GRAPE | **26.11%** | **47.14%** | **28.56%** | 27.65%| **61.10%** | **74.40%** | **44.16%**  | 15.8 h
> 100| DoGE | 24.57% | 45.33% | 27.77% | 25.35%| 59.52%| 72.20%| 42.45% | 15.0h
> 200 | GRAPE | 23.81% | 44.40% | 27.77% | **29.03%** | 59.58%| 72.70%| 42.88% | 10.3h
> 200| DoGE | 24.32% | 43.31% | 27.07% | 28.42%| 59.14%| 70.10%| 42.06% | 10.8h
>
> These results show that GRAPE consistently outperforms DoGE across all tested update frequencies. We also observe that the most frequent updates ($\Delta T=50$) do not yield the best performance. This is likely because the Rate-of-Improvement (RoI) metric becomes more reliable after the model has trained for a sufficient number of steps on the current data mixture, making overly frequent weight adjustments suboptimal.
>
> **A side note on efficiency improvement:** Regarding efficiency, while GRAPE has a theoretical FLOP overhead of ~ 28%, we note that the practical runtime is also largely influenced by the extra GPU memory occupation for storing extra gradients. A promising direction for future work is to incorporate a recently proposed "ghost gradient computation" technique [1], which could further improve the wall-clock efficiency for both GRAPE and DoGE.
>
> We will add a more comprehensive discussion on the hyperparameter tuning in the updated version of our paper.
>
> ## Q2: Benefits of the Minimax DRO Framework vs. Minimization-Only Frameworks
> The primary benefit of GRAPE's minimax (min-max) framework is that it explicitly optimizes for robustness of multi-task learning performance by adaptively tuning the task weights, while the minimization-only frameworks (like RegMix, DoGE, or Aioli) only optimizes the domain weights according to a static, pre-defined target distribution.
> Therefore, GRAPE is able to adaptively adjust the guidance distribution (i.e. task weights) for balanced and robust improvements across a diverse suite of benchmarks.
>
> ## Q3: Scalability to a large number of target tasks
> We thank the reviewer for the insightful question! The scalability of gradient-based approaches is indeed a critical issue since computing gradients for all $N$ target tasks can become a bottleneck. While our experiments used a manageable number of tasks, the GRAPE framework is flexible and can be adapted for scalability using two primary strategies:
>
> #### Method 1: Task Reweighting with Partial Feedback
> Instead of computing gradients for all $N$ tasks at every update, we can sample a small subset of $m << N$ tasks. This can be done in a principled manner using techniques from the multi-armed bandit literature (e.g., EXP3.M [2]), which balance exploiting known slow-learning tasks with exploring others. This would reduce the per-update cost from $O(N)$ to $O(m)$ without losing the core benefit of the algorithm.
>
> Denote task weights as $z(t)$ as in the paper, define a sampling distribution:
> $$p_n^{(t)} = (1-\epsilon) \cdot z_n^{(t)} + \frac{\epsilon}{N}$$
>
> to balance exploitation (focusing on tasks currently believed to be important) with exploration (ensuring all tasks remain observable).
> For each sampled task $n \in S_t$, we compute the alignment score to approximate ROI scores:
>
> $A^{(t)} = \[a^{(t)}_{1}, \hdots, a^{(t)}_{N}\]$, where $a^{(t)}_{n} = \langle \Delta_{\theta} l_n^{val}, d_t \rangle$
>
> where $\Delta_\theta l_n^{val}$ is the normalized validation gradient for task $n$, and $d_t$ is the current model update (a weighted combination of training domain gradients).
> To convert this alignment signal into a bandit-style reward in [0,1], we rescale it as follows:
> $$r_{n}^{(t)} = \frac{1}{2} (1 - a_{n}^{(t)}/C)$$
> where $C > 0$ is a scaling constant that upper-bounds the magnitude of alignment scores (e.g., a norm bound). This ensures that $r_{t,n} \in \[0,1\]$, with larger rewards assigned to more misaligned tasks—consistent with GRAPE's principle of prioritizing underperforming tasks.
> We then update the task weights using the standard multiplicative-weights rule (as in EXP3.M [2]), with importance-weighted feedback:
> $$z_{n}^{(t+1)} \leftarrow z_{n}^{(t)} · exp(\eta_{z} · r_{n}^{(t)}/p_{n}^{(t)} · 1_{n \in S_t})$$
>
> This procedure allows GRAPE to focus on the most corrective tasks **using only a small number of gradient evaluations per step**.
>
> #### Hierarchical Task Reweighting
> For corpora with thousands of fine-grained partitions, we can first group similar domains or tasks into semantically coherent clusters based on metadata or embeddings similarity. GRAPE would then operate on these high-level clusters, reweighting groups of tasks rather than individual ones. This dramatically reduces the number of entities to reweight, making the process highly scalable.
>
> [1] Data Shapley in One Training Run. Wang et. al, 2025.
>
> [2] Algorithms for adversarial bandit problems with multiple plays. T. Uchiya, A. Nakamura, and M. Kudo, 2010.

---

> ### Comment · Reviewer_qSR5 · 2025-08-02
>
> Thank you for your time! Your answers here are very interesting, and I hope they'll make it into a revision of the paper.

---

### Official Review · Reviewer_aZMB · 2025-07-01

**Clarity:** 3
**Significance:** 3
**Originality:** 4
**Rating:** 5
**Confidence:** 4

**Summary:**

This paper presents a novel method for dynamically rebalancing task weights in a bilevel optimization for data weighting where multiple tasks are simultaneously being optimized for. This approach significantly outperforms previous methods and baselines on a variety of experiments. The authors present interesting experiment setups which allow for deeper analysis of the functioning of the method. The paper concludes by presenting many compelling directions for extending this work.

**Questions:**

The authors present an approach where a first-order taylor expansion is used to approximate RoI in order to provide the setup for applying GroupDRO. Did the authors impelment the simpler version of just calculating RoI directly per-step and updating the task-weights via a simple heuristic (ie using the softmax of the inverse RoIs)? This seems a simple and efficient baseline that would be useful to compare against before getting more complicated.

In section 4, much space is spent on comparing GRAPE to GRAPE-ema and GRAPE-gap (l 285-314, fig 6, table 2). This is fine to establish that GRAPE is the better of the three methods but not terribly interesting beyond that. Relatively little space is spent on the next section (l 315-326) discussing the relative performance of GRAPE under different task distribution settings, where the results are referred to in the appendix. Is this not critical information to understand both how the method is functioning and in what settings it might be useful to be applied? Consider showing less of the GRAPE-ema comparison and more of the GRAPE-in-different-task-settings.

Did the authors run any experiments where the relative improvements between tasks are not uniformly weighted? In practical terms, practitioners may prioritize some tasks over others in their downstream evaluations, so it would be very useful and interesting to know if the strong performance of GRAPE extended to these settings. Do the authors have any comments / intuitions on this?

Nits:
line 288:  typo on "improvement"

**Ethical Concerns:**

["NO or VERY MINOR ethics concerns only"]

**Limitations:**

Yes.

**Quality:**

4

**Strengths And Weaknesses:**

Strengths:
The paper is clearly written, the algorithm presented is a novel approach to the problem. The results are very compelling. The presented analyses are very thorough and provide fascinating insight into the function of the method. The result is very likely to be used by the research community in future work.

Weaknesses:

magical hyperparams:
(This is bit of a nit, and does not detract terribly from the overall high quality of this work)
It makes sense that ∆T_z and ∆T_alpha should not be 1, for both computational cost and diminishing retuns on frequency of update. It would be very useful however to have some sense of what this tradeoff looks like in practice rather than simply punting the hyper-param tuning (lines 328-334). It does not need to be exhaustive, but it would be nice to include *some* analysis of this tradeoff. Did the authors ever run any exp with ∆T == 50? == 200? A simple sweep on a tiny model would suffice.

---

> ### Author Rebuttal · Authors · 2025-07-31
>
> We are grateful to the reviewer for their careful assessment of our work and the insightful feedback. We agree with the suggested points and provide our responses below:
>
> ## Q1: Hyperparameter tuning & Runtime:
> We perform a hyperparameter sweep on the reweighting intervals of $\Delta T_{\alpha}, \Delta T_{z} = 50, 100, 200$ for both GRAPE and DoGE. We train 125M model for 50k steps and report the final task accuracies and run time as follows:
>
> $\Delta Τ_{\alpha}, \Delta Τ_{z}$ | Method | ARC-C | ARC-E | Hellaswag | LogiQA | PIQA | SciQ | Average | Runtime (H100)
> ----|--------|---------------|----------|-----------|--------|-------|-------|---------|--------
> 50 | GRAPE | 25.09% | 45.83% | 27.86% | 25.50%| 60.12%| 73.30%| 42.95% | 23.9h
> 50 | DoGE | 24.40% | 43.64% | 27.53% | 26.88%| 58.98%| 70.10%| 41.92% | 22.3h
> 100 | GRAPE | **26.11%** | **47.14%** | **28.56%** | 27.65%| **61.10%** | **74.40%** | **44.16%**  | 15.8 h
> 100 | DoGE | 24.57% | 45.33% | 27.77% | 25.35%| 59.52%| 72.20%| 42.45% | 15.0h
> 200 | GRAPE | 23.81% | 44.40% | 27.77% | **29.03%** | 59.58%| 72.70%| 42.88% | 10.3h
> 200 | DoGE | 24.32% | 43.31% | 27.07% | 28.42%| 59.14%| 70.10%| 42.06% | 10.8h
>
> These results show that GRAPE consistently outperforms DoGE across all tested update frequencies. We also observe that the most frequent updates ($\Delta T=50$) do not yield the best performance. This is likely because the Rate-of-Improvement (RoI) metric becomes more reliable after the model has trained for a sufficient number of steps on the current data mixture, making overly frequent weight adjustments sub-optimal.
>
> **A side note on efficiency improvement:** Regarding efficiency, while GRAPE has a theoretical FLOP overhead of ~ 28%, we note that the practical runtime is also largely influenced by the extra GPU memory occupation for storing extra gradients. A promising direction for future work is to incorporate a recently proposed "ghost gradient computation" technique [1], which could further improve the wall-clock efficiency for both GRAPE and DoGE.
>
> We will add a discussion on the hyperparameter tuning in the updated version of our paper.
>
>
> ## Q2: GRAPE with exact ROI scores
> We intended to predict the Ratio-of-Improvements at the next step using a look-ahead strategy, where the first-order approximation stands. Also, the gradient alignment between tasks and training set can provide more information about the optimization directions on the landscape than the exact ROI score.
>
> To validate this, we ran an experiment with the exact ROI scores, where we calculated ROI directly to update task weights at every reweighting step and then update domain weights according to the task weights. The 5-shot accuracy on the target tasks are presented as follows:
>
> Method | ARC-C | ARC-E | Hellaswag | LogiQA | PIQA | SciQ | Avg.
> ----|--------|---------------|----------|-----------|--------|-------|-------
> Uniform | 23.12% | 40.74% | 27.02% | 27.96%| 58.98%| 67.50%| 40.89%
> GRAPE | 26.11% | 47.14% | 28.56% | 27.65%| 61.10%|74.40%| 44.16%
> DoGE | 24.57% | 45.33% | 27.77% | 25.35%| 59.52%| 72.20%| 42.45%
> GRAPE+direct ROI | 23.29% | 42.76% | 27.16% | 27.50% | 56.86%| 72.80% | 41.73%
>
> The results above show that the underlying GroupDRO framework in GRAPE is more effective than the directly computed ROI scores. We will add more discussions and analysis on this ablation in the revised version of the paper.
>
> ## Q3: Apply on tasks with priority preference
> GRAPE is primarily designed for robustness, where the goal is to achieve balanced performance by automatically identifying and lifting the lagged tasks.
> In practice, we can run GRAPE with task-weights initialized non-uniformly according to the pre-defined task priority. However, the principle of GRAPE that continuously optimizes the worst-improved task could conflict with the predefined priority-order. For instance, GRAPE might upweight a low-priority task if it falls significantly behind, which would counter the user's explicit preference.
>
> Therefore, if a practitioner has pre-defined, static priorities for tasks, applying DoGE with a static non-uniform task-weights would be more appropriate.
>
> [1] Data Shapley in One Training Run. Wang et. al, 2025.

---

### Official Review · Reviewer_P62V · 2025-07-03

**Clarity:** 3
**Significance:** 3
**Originality:** 3
**Rating:** 5
**Confidence:** 3

**Summary:**

This paper docuses on optimizing data mixtures in pretraining language models with the goal of improving performance of the models on multiple target tasks at the same time. This is an extension of past research which set this up mostly with the goal of optimizing perormance on a single target task.

The paper introduces a method to adjust dynamically both sample weights from source domains, as well as task weights. Adjusting task weights dynamically allows the model to prioritize different tasks at different times in training. The learning problem is formalzied as a minimax optimization problem, where the task weights are selected using group distributed-robest-optimization and the domain weights are maximizing loss reduction on the weighted tasks.

The experiments are conducted on two setups:
1. learning from ClimbLab and SlimPajama and testing on a set of 6 reasoning dataset using 5-shot in-context learning
2. learning from Wikipedia data from 6 languages and evaluating using perplexity on 8 other languages

**Questions:**

-  baselines would require a bit more description for example it is not clear what the domain weights are trained and their frequency in regmix

- line 244:  Ukrian > Ukrainian

**Ethical Concerns:**

["NO or VERY MINOR ethics concerns only"]

**Final Justification:**

Thank you for the additional clarifications. These were relatively minor points, and the response will improve the revision. I remain positive overall on the paper.

**Limitations:**

Highlighting these limitations of the research, but I acknowledge the space and scope limitations of the current paper:

- There is still to be studies what is happening after running SFT/RL on top of these trained models
- Models trained are only up to 0.75 on limited training steps
- Comprehensive baselines, but only for the first experiment; this is probably fine given that DoGE is the best non-GRAPE method in that experiment. But for completeness, it would be good to have more baselines for the other experiments

**Paper Formatting Concerns:**

No concerns

**Quality:**

3

**Strengths And Weaknesses:**

Strenghts:
- new problem/experimental setup (multi-target testing)
- theoretical justification of the algorithm
- comparison with many baselines for the first experiment, including all natural ones from past work
- experimentation on 2 different setups
- feature analysis and interpretation of results

Weaknesses:
- evaluation for the multi-lingual setup is only done using perplexity, which may not be related to actual task performance. One could have evaluated instead on downstream tasks with ICL (e.g. NER or classification) or machine translation. These would make these results stronger

---

> ### Author Rebuttal · Authors · 2025-07-31
>
> We thank the reviewer for the thoughtful questions and constructive suggestions! We provide our response as follows:
> ## Q1: Clarification on the baselines
> We provide detailed implementation descriptions for all baseline methods in **Appendix D.1** (line 620-674). We also provided the distribution of regmix-optimized domain weights in Figure 7,8,11,12,16,17. The actual domain weights from RegMix on the CLIMBLAB dataset with 6 target tasks is presented as follows:
>
> | C1   | C2   | C3  | C4  | C5  | C6 | C7 | C8 | C9 | C10 |
> |-------|-----|------|-------|------|-----|-----|-----|----|------|
> |0.00 | 0.01 |0.00 |0.19 |0.01 |0.03 |0.02| 0.02 | 0.01 | 0.00 |
> |C11 | C12  | C13 |C14 | C15 |C16 | C17  | C18  | C19 | C20 |
> |0.01 | 0.02 | 0.00 |0.00 |0.02 |0.01| 0.62 | 0.02| 0.00 |0.00|
>
> We will add the tables with the actual domain weights from all baseline methods in our revised version for better readability and clarity!
>
>
> ## Q2: Performance after SFT
> To investigate if GRAPE's pretraining advantages remains after post-training, we conducted supervised fine-tuning experiments on the pretrained 125M models.
> We present the 5-shot accuracy scores before and after supervised fine-tuning on PIQA, SciQ, ARC-Easy, ARC-Challenge as follows:
>
> | Method | Base | Base + SFT | DOGE | DOGE + SFT | GRAPE | GRAPE + SFT |
> |--------|------|-------------------|------|-------------------|-------|-------------------|
> | PIQA | 58.98| 63.06 | 59.52| 62.62 | 61.10 | **64.25** |
> | SciQ | 67.50| 75.3 | 72.20| 79.0 | 74.40 | **79.70** |
> |ARC-E| 40.74| 44.23 | 45.33 | 48.44|47.14| **48.86**|
> |ARC-C | 23.12 |  **27.22** | 24.57 | 26.54 | 26.11 | 27.13|
>
> The results show that GRAPE keeps the advantage after the SFT stage in most of the tasks. On ARC-Challenge, GRAPE outperforms DoGE and performs comparably as the uniform baseline.
> We did not conduct the RL here since we did not apply a relevant task as our target, but how to select the optimal data for RL is definitely a critical and challenging direction to explore.
>
> ## Q3: Limited training length and baselines in large-scale experiments
> Due to the limitation of computation budget, we did not manage to pretrain models larger than 0.7B on sufficient numbers of tokens.
> As the reviewer notes, we prioritized comparing against DoGE as it was the strongest baseline under the limited resources but did not include RegMix and CRISP baselines in 0.7B experiments. We will be sure to add the results for RegMix and CRISP to the 0.7B experiments in the final version of the paper to ensure a complete comparison.

---

### Official Review · Reviewer_yAw7 · 2025-07-10

**Clarity:** 2
**Significance:** 3
**Originality:** 3
**Rating:** 4
**Confidence:** 4

**Summary:**

This paper considers the problem of choosing the best pretraining data mixture that result in good accuracy on a set of downstream target tasks. Specifically, the authors propose optimizing over the relative proportions/weights of a set of pretraining domain to minimize the worst case task (GRAPE). Group distributionally robust optimization (DRO) is applied to choose domain weights that result in as much (normalized) loss reduction as possible. Experiments on training language models for reasoning tasks and multilingual capabilities show improvements over baselines.

**Questions:**

- line 100: why is taking a small learning rate justified (in order to take the Taylor expansion)?
 - line 288: typo "improvmenet"
 - Theorem 2.1: this theorem seems to prove convergence to a global optimum, but this shouldn't be true for general (non-convex) loss functions?
 - Theorem 2.2: assuming that the task losses are strongly convex is a pretty strong/unrealistic assumption, but is this necessary, given that monotonicity if more of a local notion?
 - Given that the method is about minimizing the worst case loss, why is average accuracy what is mainly reported?
 - From 125M to 0.7B parameters, it seems like the improvement of GRAPE over the baseline is reduced; what are the prospects of GRAPE still outperforming the baseline when scaled up to 7B or beyond?
 - Can you comment on the runtime required of the different methods?
 - How well does GRAPE do compared to the oracle of selecting the best data for a given task? I'd like to understand how much of the value is being able to select data period versus being able to juggle multiple tasks.

**Ethical Concerns:**

["NO or VERY MINOR ethics concerns only"]

**Limitations:**

yes

**Quality:**

3

**Strengths And Weaknesses:**

- Overall, the problem (of finding good pretraining data for downstream tasks) is well-motivated. The paper proposes a new solution that leverages principles from robust optimization (building off of FAMO, which updates parameters to reduce task loss) and appears to work quite well.
 - The paper tries to upweight tasks that are improving more slowly, but I wonder if this is the right principle; if you have a task that has high error and cannot be improved, it seems like this will be maximally upweighted, which seems not ideal. Group DRO methods sometimes use a reference model that tries to take this into account; it would be nice to discuss this more.
 - Appendix D.1 says that DoGE used the same hyperparameters as GRAPE, but is there any reason to believe these are optimal? In general, for fair comparison, the hyperparameters need to be tuned carefully and separately for each method.

---

> ### Author Rebuttal · Authors · 2025-07-31
>
> We thank the reviewer for the effort to read through our work and the constructive feedback! These questions have helped us clarify key aspects of our work. We provide our responses as follows:
> ## Q1: Hyperparameter tuning & Runtime:
> We perform a hyperparameter search on the reweighting intervals $\Delta T_{\alpha}, \Delta T_{z} = 50, 100, 200$ for both GRAPE and DoGE. We train 125M model for 50k steps and report the final task accuracies and run time as follows:
>
> $\Delta Τ_{\alpha}, \Delta Τ_{z}$ | Method | arc_challenge | arc_easy | hellaswag | logiqa | piqa | sciq | Average | Runtime (H100)
> ----|--------|---------------|----------|-----------|--------|-------|-------|---------|--------
> 50 | GRAPE | 25.09% | 45.83% | 27.86% | 25.50%| 60.12%| 73.30%| 42.95% | 23.9h
>  50 | DoGE | 24.40% | 43.64% | 27.53% | 26.88%| 58.98%| 70.10%| 41.92% | 22.3h
> 100 | GRAPE | **26.11%** | **47.14%** | **28.56%** | 27.65%| **61.10%** | **74.40%** | **44.16%**  | 15.8 h
> 100 | DoGE | 24.57% | 45.33% | 27.77% | 25.35%| 59.52%| 72.20%| 42.45% | 15.0 h
> 200 | GRAPE | 23.81% | 44.40% | 27.77% | **29.03%** | 59.58%| 72.70%| 42.88% | 10.3h
>  200| DoGE | 24.32% | 43.31% | 27.07% | 28.42%| 59.14%| 70.10%| 42.06% | 10.8h
>
>
> These results show that GRAPE consistently outperforms DoGE across all tested update frequencies. We also observe that the most frequent updates ($\Delta T$=50) do not yield the best performance. This is likely because the Rate-of-Improvement (RoI) metric becomes more reliable after the model has trained for a sufficient number of steps on the current data mixture, making overly frequent weight adjustments sub-optimal.
>
> A side note on efficiency improvement: Regarding efficiency, while GRAPE has a theoretical FLOP overhead of ~ 28%, we note that the practical runtime is also largely influenced by the extra GPU memory occupation for storing extra gradients. A promising direction for future work is to incorporate a recently proposed "ghost gradient computation" technique [1], which could further improve the wall-clock efficiency for both GRAPE and DoGE.
>
> We will add a discussion on the hyperparameter tuning in the updated version of our paper.
>
> ## Q2: Average versus worst-case accuracy
> We did not report worst-case accuracy as it can be volatile and skewed by a single, exceptionally difficult task, which may not reflect the model's broader capabilities across a diverse task suite.
>
> However, to directly evaluate the worst-case performance against our optimization objective (cross-entropy loss), we provide extensive results on worst-case log-perplexity in Table 5, Table 8, and Table 9. The results demonstrate that GRAPE consistently achieves superior (lower) worst-case perplexity compared to baselines across numerous task configurations.
>
> ## Q3: Justification on the small learning rate assumption
> The assumption of a small learning rate is a standard condition for applying the first-order Taylor expansion to approximate the one-step loss improvement, which is a common practice in the theoretical analysis of gradient-based algorithms. Empirically, it can be justified as the learning rates used in large-scale language model pretraining are typically considered small relative to the optimization landscape.
>
> ## Q4: Assumptions on Theorems
> - **On Theorem 2.1**:  we agree with the reviewer that the general LLM training is a highly non-convex scenario. However, the convex assumption is commonly used for theoretical analysis for optimization to provide theoretical grounding and intuition for the algorithm's behavior in an idealized setting, such as the very first work on influence function [2]. Our empirical results show that even in the non-convex LLM pretraining practice, GRAPE with the underlying group-DRO algorithm can still provide valuable insight on the data mixture selection and brings significant empirical improvements.
> - **On Theorem 2.2**: We thank the reviewer for this sharp observation. We agree that the μ-strongly convex condition is not necessary to prove monotonic variance reduction. We will remove this assumption in the revised version of our paper.
>
> ## Q5: Scaling behaviors of GRAPE
> We thank the reviewer for this insightful question! While GRAPE outperforms all baselines at both the 125M and 0.7B scales, the relative improvement over the uniform baseline is less pronounced for the larger model. We hypothesize a few reasons:
> - (1) **Limited Training Tokens**: due to the limited computation budget, The 0.7B model was trained on 13B tokens, which may be insufficient to show the full benefits of a dynamic data curriculum at that scale;
> - (2) **Sub-optimal Hyperparameters**: The optimal hyperparameters may differ across model scales, and our settings may perform better on 125M models than 0.7B models;
> - (3) **Shifting Task Dynamics**: while the model is scaling up, the conflict pattern among various target tasks can be shifted. For example, larger models can be more sensitive to task conflicts or easier to generalize across various domains. Understanding the scaling effect of the multi-task learning behavior is an intriguing direction to explore. We will leave it to our future work.
>
> ## Q6: Comparison to the oracle of data selection
> While the true oracle of data mixture is hard to define in the context of LLM pretraining, we apply CRISP as a proxy for a static oracle baseline. CRISP identifies the domain sampling weights as the closest training data distribution to the target data distribution according to the sequence embedding similarity. Therefore, the CRISP mixture is determined using an auxiliary embedding model (e.g. sentence-bert) before the training begins.
> Our results show that GRAPE's dynamic approach significantly outperforms this static oracle. In Table 1,5,6,7,8, GRAPE largely improves perplexity and reasoning accuracies compared to CRISP; according to Figure 19,20, in the multilingual setting, the static mixtures from CRISP can sabotage performance on most languages, whereas GRAPE delivers consistent gains on all the target languages, leveraging the dynamic data curriculum.
>
> [1] Data Shapley in One Training Run. Wang et. al, 2025.
>
> [2] Understanding Black-box Predictions via Influence Functions. Koh and Liang, 2018.

---

> > ### Comment · Reviewer_yAw7 · 2025-08-08
> >
> > Thank you for the response to my questions.  I'd be happy to see this paper get in.

---

### Decision · Program_Chairs · 2025-09-17

**Decision:**

Accept (poster)

**Comment:**

The paper introduces a new method called GRAPE for data mixing for pretraining large language models. The main problem it addresses is that existing techniques often optimize the pretraining data for a single goal/metric, which can hurt the model's performance on other tasks. GRAPE is designed to improve a model's performance across multiple different tasks at the same time. It works by dynamically adjusting two things during training: the mix of the source data and the importance given to each target task. The system identifies tasks that the model is struggling to learn and prioritizes them by giving them higher weight. Then, it adjusts the data mixture to specifically help improve performance on these prioritized, harder tasks. Reviewers agreed that this was a well-motivated and novel extension of previous work, with strong experimental results.

The reviewers' feedback was mostly positive, leading to three "Accept" and two "Borderline Accept". Initially, several issues raised including the practicality of the method, such as its runtime cost, the difficulty of tuning its many hyperparameters, and whether its benefits would diminish as models get larger. Reviewers also questioned aspects of the evaluation, such as the fairness of hyperparameter settings for baseline methods and the scope of the test sets. Despite these initial concerns/questions, the authors' responses addressed most of these concerns and overall, the reviewers concluded that the paper made a meaningful contribution to the field. The authors should include the additional experiments and discussions in the revised paper.